# PosterAgent: Agentic Poster Generation via Stage-Aware Reinforcement Learning

**Zhuocheng Yu** [* 1 2]   **Feng Zhang** [3]   **Sujian Li** [1 2]   **Kai Jia** [3]

## Abstract

Poster generation is a complex task demanding a harmonious integration of visual aesthetics and information hierarchy. While recent text-to-image models have advanced visual synthesis, they remain non-editable and struggle with precise text rendering. Conversely, existing layout-generation methods offer structure but typically rely on static, one-shot predictions, lacking a mechanism for self-correction essential to professional design. Inspired by the iterative workflow of human designers, we introduce **PosterAgent**, a novel framework that reformulates poster creation as an agentic workflow involving initial drafting followed by iterative refinement. To effectively train this multi-turn capability, we propose **Stage-Aware Reinforcement Learning (SARL)**, which decouples the optimization into draft-specific and refinement-specific phases, ensuring precise credit assignment for both initial drafting and incremental refinement actions. Extensive experiments demonstrate that PosterAgent significantly outperforms strong baselines, validating the potential of agentic systems in graphic design.

## 1. Introduction

Posters are a high-impact medium for visual communication: they distill complex ideas into a single glanceable artifact and are pervasive in advertising, events, and academic dissemination. Yet producing a "good" poster remains notoriously difficult—it demands a harmonious integration of visual aesthetics, information hierarchy, and deliberate layout planning. This practical importance and difficulty have spurred growing research interest in automating layout design (Yamaguchi, 2021; Cheng et al., 2025) and poster generation (Inoue et al., 2023; Hu et al., 2025), with the aim of allowing non-expert users to transform abstract concepts into visually compelling posters.

With the rapid progress of diffusion-based text-to-image (T2I) models (Wu et al., 2025a; Cao et al., 2025; Seedream et al., 2025), a natural approach to poster generation is to directly synthesize visuals from text prompts. However, such methods face critical limitations for design tasks: they sometimes render text inaccurately, especially in text-dense scenes (Wang et al., 2025a) and produce non-editable raster outputs that hinder users' further adjustments (Zhang et al., 2025). To address these issues, recent research has shifted toward layout-driven approaches that first generate a structured blueprint (e.g., specifying text boxes, images, and other design elements) before rendering the final poster (Jia et al., 2023; Inoue et al., 2024; Zhang et al., 2025; Wang et al., 2025a). While this paradigm improves controllability, it still exhibits notable shortcomings. These methods typically rely on supervised fine-tuning on specific graphic datasets, which can limit adaptability to new domains or styles (Chu et al., 2025; Wu et al., 2025b). More importantly, both paradigms share a core weakness: they operate as one-shot generators with no capacity for self-critique or iterative refinement. Consequently, the models cannot receive feedback from their own outputs, identify suboptimal choices, or perform purposeful edits in response to their previous designs.

Inspired by how human designers work, we argue that poster creation is inherently an *agentic* process rather than a one-shot generation problem. Designers do not directly "render" a final poster from a textual brief; instead, they iteratively (1) interpret the intent and plan an information hierarchy, (2) externalize a draft layout in an editable form, (3) render it to see how it reads visually, and (4) revise specific parts based on the observed outcome. This workflow aligns naturally with the emerging agentic paradigm (Yao et al., 2022; Yang et al., 2024; Wang et al., 2024; Li et al., 2025b), which frames problem solving as sequential decision making: the agent maintains an evolving state, takes actions,

*Work done during internship at BandAI, ByteDance [1]School of Computer Science, Peking University, Beijing, China [2]National Key Laboratory for Multimedia Information Processing, Peking University, Beijing, China [3]ByteDance, Beijing, China. Correspondence to: Sujian Li <lisujian@pku.edu.cn>.

receives feedback from the environment, and iteratively updates its decisions through self-reflection to better achieve the objective.

Motivated by this perspective, we introduce **PosterAgent**, a novel multi-turn agentic framework that models poster creation as an iterative workflow, powered by a Multimodal Large Language Model (Bai et al., 2025; Wang et al., 2025b; Li et al., 2025a; Guo et al., 2025). Given a user's textual instruction, PosterAgent first translates it into a structured JSON design. A renderer then converts this specification into a visual poster. Crucially, the agent is subsequently presented with its own initial design and the resulting poster, prompting it to act as its own critic, reflect on potential flaws and identify possible enhancements, and produce a modified design. This loop can continue for multiple turns, enabling progressive optimization.

To fully realize this paradigm's potential, we employ reinforcement learning (Kaelbling et al., 1996) to train the agent. However, training an iterative designer exposes a key weakness of the common outcome-only reward practice (Ouyang et al., 2022; Shao et al., 2024): assigning a single score to the final poster blurs two distinct behaviors—initial drafting and subsequent self-refinement. A high final score may simply come from an already-strong initial draft, while a low final score may be caused by a weak initial draft even if the refinement steps made meaningful improvements. This yields a noisy and poorly attributed learning signal for refinement. To address this, we propose **Stage-Aware Reinforcement Learning (SARL)**, a novel RL framework that decouples the rollout-and-scoring procedure in each training step into two phases. Concretely, during each training step, we employ a two-phase sampling strategy to construct stage-specific training batches. First, we sample independent initial drafts to create a draft group, evaluating them against a quality rubric. Second, to strictly isolate the impact of editing decisions, we fix a single starting draft from the first phase and branch multiple parallel self-revision trajectories from it to form a refinement group, which is scored by a relative improvement judge. We then compute group-wise normalized advantages and jointly update the policy. This mechanism ensures that the draft reward specifically targets layout planning, while the refinement reward captures the marginal value of editing actions without being confounded by the variance of the starting states.

Our contributions can be summarized as follows:

- We propose PosterAgent, a novel agentic framework that models poster creation as an iterative workflow, enabling both initial design generation and self-refinement.
- We design SARL, a new reinforcement learning paradigm that decouples and jointly optimizes the agent's drafting and refinement abilities with stage-specific rewards, enabling more precise credit assignment.

- Through extensive experiments, we show that PosterAgent trained with SARL outperforms strong baselines, and that iterative self-refinement consistently improves quality and editability, with ablations validating each component.

**Conflict of Interest Disclosure**   Authors F. Zhang and K. Jia are employed by ByteDance, which leads the development of Seedream 3.0 and Seedream 4.0, two of the text-to-image models evaluated as baselines in this paper. Seedream 4.0 is additionally used as the image generator within our rendering environment.

## 2. Related Work

**Automated Layout and Graphic Design**   Research in automated graphic design has evolved from heuristic constraint satisfaction (O'Donovan et al., 2014) to deep generative models that learn layout distributions from data. Early learning-based approaches utilized GANs (Li et al., 2019; Kikuchi et al., 2021) and VAEs (Yamaguchi, 2021; Jyothi et al., 2019) to synthesize spatial arrangements of graphical elements, while subsequent works adopted Transformer-based architectures to capture complex element relationships (Gupta et al., 2021; Arroyo et al., 2021). More recently, the focus has shifted toward holistic systems that generate both layout and content. Notable examples include COLE (Jia et al., 2023) and OpenCOLE (Inoue et al., 2024), which employ hierarchical diffusion models to render editable layers, and recent multimodal approaches that integrate retrieval with generation (Hu et al., 2025; Cheng et al., 2025). Despite these advancements, prior methods typically lack the mechanism to revisit or iteratively repair a design based on visual rendering artifacts, a gap our work addresses by introducing an agentic feedback loop.

**LLM-based Agents**   Large Language Models have demonstrated remarkable capabilities as autonomous agents capable of planning and tool use (Xi et al., 2025; Wang et al., 2024). A cornerstone of this success is the "reasoning-through-acting" paradigm, exemplified by frameworks such as ReAct (Yao et al., 2022) and intricate multi-agent collaborations (Hong et al., 2023). Crucially, recent work highlights the efficacy of intrinsic self-correction, where models improve outputs through iterative critique (Shinn et al., 2023; Madaan et al., 2023). While these refinement strategies have been successfully applied to code generation (Yang et al., 2024) and mathematical reasoning (Shao et al., 2025), their application to visual design remains underexplored. Unlike text-based tasks where feedback is often symbolic, graphic design requires interpreting pixel-level feedback against aesthetic constraints. Our work extends agentic self-refinement to the multimodal domain, establishing a visual feedback loop where the agent iteratively critiques and modifies the rendered artifacts to achieve professional design standards.

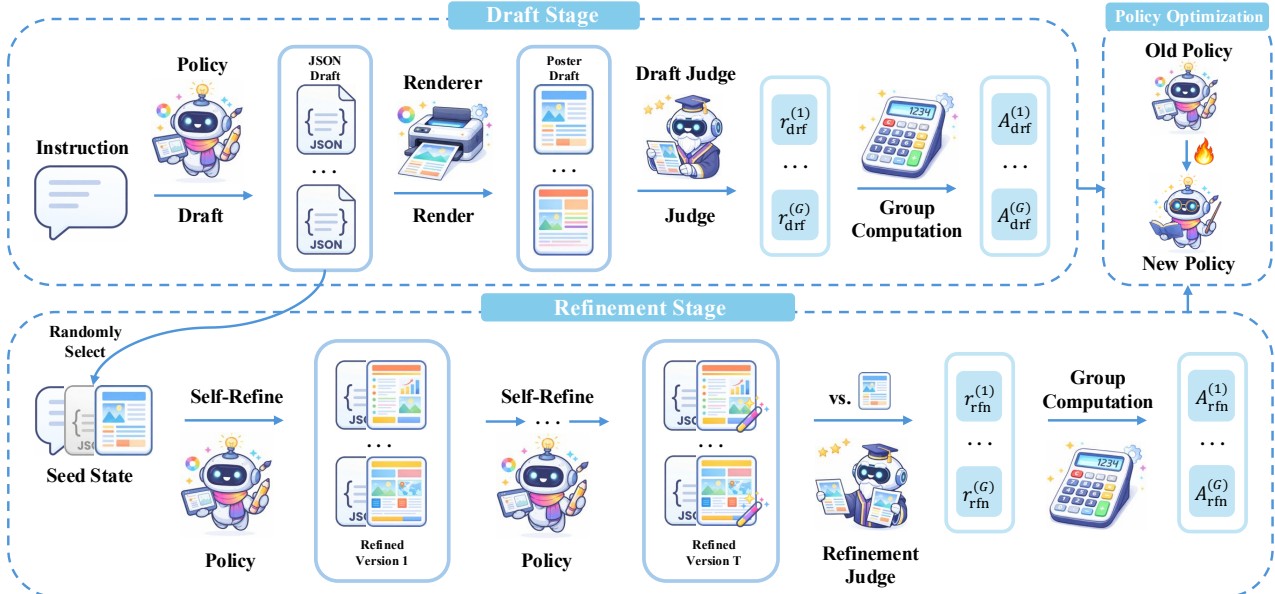

*Figure 1.* Overview of Stage-Aware Reinforcement Learning (SARL). The framework employs a two-phase training strategy: (top) the Draft Stage samples multiple initial designs and evaluates their absolute quality, while (bottom) the Refinement Stage branches from a fixed seed state to assess relative improvements. The trajectories collected from the two stages are then used together to optimize the policy.

## 3. Methodology

In this section, we present the proposed framework for automated poster generation. We first present PosterAgent, our proposed agentic framework that treats poster generation as a sequential decision-making process involving design, rendering, and reflection. Subsequently, we introduce our novel training paradigm, Stage-Aware Reinforcement Learning (SARL), which decouples the optimization of drafting and refinement capabilities to address the credit assignment challenges in iterative design tasks.

### 3.1. The PosterAgent Framework: An Iterative Design Agent

Unlike traditional approaches that treat poster generation as a one-shot conditional generation task (i.e., mapping a text instruction directly to an image), we formulate the problem as a Partially Observable Markov Decision Process (POMDP), where the agent (an MLLM) interacts with a rendering environment to progressively refine a poster design.

#### 3.1.1. ACTION SPACE: STRUCTURED DESIGN SPECIFICATION

The action space is defined as the set of all valid JSON-based design specifications. An action $a_t$ at time step $t$ represents a complete description of the poster layout and content in JSON format.[1] We abstract $a_t$ as a collection of discrete graphical elements:

$$a_t = \{e_1, e_2, \ldots, e_{N_t}\}, \tag{1}$$

where $N_t$ denotes the number of graphical elements. Each element $e_i$ can be formulated as a tuple consisting of its type (e.g., text, image, shape, etc.) and a type-specific attribute set. Our framework supports essential element types for poster design, each with a corresponding attribute schema. Detailed specifications are provided in Appendix D.1.

#### 3.1.2. OBSERVATION SPACE: RENDERED POSTER WITH DIAGNOSTIC FEEDBACK

The environment consists of a deterministic renderer $\mathcal{M}$ that converts the structured action $a_t$ into a multimodal observation:

$$o_t = \mathcal{M}(a_t) = (I_t, W_t), \tag{2}$$

where $I_t$ is the rendered poster image and $W_t$ is a set of execution warnings.

**Visual Observation** The image $I_t$ is the rasterized realization of the structured design. This is the key perceptual signal for the agent's reflection: it exposes typography readability, visual hierarchy, spacing, and overall composition

---

[1]In practice, the policy generates a chain-of-thought reasoning trace before outputting the JSON design specification. We denote only the structured JSON output as the action $a_t$ for clarity.

— properties that are hard to reliably infer from the JSON alone.

**Textual Observation** The renderer also performs sanity checks on the layout, returning specific warnings for issues such as text overflow (where the chosen font size causes the text to overflow the bounding box) or invalid geometric parameters. We view these warnings as a form of symbolic feedback that complements visual observation, enabling targeted fixes in the subsequent action.

### 3.1.3. ITERATIVE DESIGN PROCESS: DRAFTING AND REFINEMENT IN A UNIFIED POMDP

PosterAgent follows a multi-turn interaction loop under a single POMDP formulation. At each iteration $t$, the policy $\pi_\theta$ conditions on the agent state $s_t$ and outputs a complete JSON design specification:

$$a_t \sim \pi_\theta(\cdot \mid s_t), \tag{3}$$

where the agent state $s_t$ aggregates the user instruction $u$ and the history of past interactions:

$$s_t = (u, a_0, o_0, a_1, o_1, \ldots, a_{t-1}, o_{t-1}). \tag{4}$$

Note that although every action $a_t$ lives in the same action space, the *role* of the action differs across turns: $a_0$ is the initial *draft* generated from the user instruction, while $\{a_t\}_{t \geq 1}$ are *refinement* actions that revise the previous design in response to what the agent observes from its own rendered outcome. In this work, we follow a fixed-horizon setting where the refinement process terminates after a predefined number of steps $T$, which facilitates stable RL training and consistent evaluation.

### 3.2. Stage-Aware Reinforcement Learning (SARL)

#### 3.2.1. MOTIVATION: WHY A SINGLE OUTCOME REWARD IS INADEQUATE

A prevalent practice in RL fine-tuning for generative models is to assign a single scalar reward to the final output (i.e., an *outcome-only* reward) and use it as the *shared* training signal for *all* intermediate decisions along the rollout. In our setting, a rollout for instruction $u$ produces an initial draft $a_0$, then iteratively refines it into $\{a_t\}_{t=1}^T$, yielding the rendered posters $\{I_t\}_{t=0}^T$. Outcome-only optimization uses a reward of the form:

$$r_{\text{out}} = \mathcal{R}_{\text{out}}(u, I_T), \tag{5}$$

where $\mathcal{R}_{\text{out}}$ is an evaluator of the final poster. However, $r_{\text{out}}$ entangles *two qualitatively different behaviors*: (1) the ability to propose a strong *initial draft* (good hierarchy and layout planning), and (2) the ability to perform effective

*self-refinement* (targeted edits that improve a given draft). A high $r_{\text{out}}$ may arise from an already-strong $a_0$ with minimal refinement, while a low $r_{\text{out}}$ can be caused by a weak $a_0$ even if the refinement steps are beneficial. This confounding yields noisy credit assignment for refinement decisions.

To address this, we introduce **Stage-Aware Reinforcement Learning (SARL)**, which explicitly separates the learning signal for drafting and refinement via two stage-specific rewards and a two-phase sampling strategy per training step.

#### 3.2.2. REWARD DESIGN

Poster design lacks a single canonical "correct answer", making automatic reward construction non-trivial. Training a dedicated reward model would additionally require curated preference data and incur non-negligible training cost. We therefore adopt an MLLM-as-Judge scheme: a strong black-box MLLM is prompted to score the generated posters, producing rewards used for RL optimization. We define two rewards:

**Draft Reward** Given instruction $u$ and the rendered initial draft $I_0$, the MLLM judge outputs:

$$r_{\text{drf}} = \mathcal{R}_{\text{drf}}(u, I_0). \tag{6}$$

$\mathcal{R}_{\text{drf}}$ assesses the *absolute* quality of the initial draft along several rubric dimensions, then averages the scores across dimensions as the draft reward.

**Refinement Reward** To specifically train the agent's self-refinement ability, we use a relative improvement judge. Given $(u, I_0, I_T)$, where $I_T$ is the refined poster after $T$ refinement turns, the judge outputs

$$r_{\text{rfn}} = \mathcal{R}_{\text{rfn}}(u, I_0, I_T). \tag{7}$$

$\mathcal{R}_{\text{rfn}}$ compares $(I_0, I_T)$ under the same instruction and quantifies how much the refinement process improves the poster.

Finally, both rewards are normalized to $[0, 1]$ before advantage computation. Full judge prompts and rubrics are provided in Appendix D.2.

#### 3.2.3. STAGE-AWARE SAMPLING AND SCORING

During each training step, for a given user instruction $u$, we implement the two-stage sampling and scoring strategy to construct two stage-specific trajectory groups for updating the policy.

**Stage 1: Draft** We perform $G$ independent rollouts of the initial draft action from the policy $\pi_{\theta_{\text{old}}}$ conditioned on $u$,

and then render each draft with the renderer:

$$\left\{ a_0^{(i)} \right\}_{i=1}^{G} \sim \pi_{\theta_{\text{old}}} ( \cdot \mid u ), \tag{8}$$

$$\left\{ o_0^{(i)} \right\}_{i=1}^{G} = \left\{ \mathcal{M} \left( a_0^{(i)} \right) \right\}_{i=1}^{G}$$

$$= \left\{ \left( I_0^{(i)}, W_0^{(i)} \right) \right\}_{i=1}^{G}. \tag{9}$$

This yields the group of draft trajectories:

$$\mathcal{G}_{\text{drf}} = \left\{ \tau_{\text{drf}}^{(i)} \right\}_{i=1}^{G} = \left\{ \left( u, a_0^{(i)}, o_0^{(i)} \right) \right\}_{i=1}^{G}. \tag{10}$$

Each trajectory in this group is evaluated using the draft judge model:

$$\mathbf{r}_{\text{drf}} = \left\{ r_{\text{drf}}^{(i)} \right\}_{i=1}^{G} = \left\{ \mathcal{R}_{\text{drf}} \left( u, I_0^{(i)} \right) \right\}_{i=1}^{G}. \tag{11}$$

**Stage 2: Iterative Self-Refine** To fairly compare the effects of the refinement process, we randomly select a single trajectory $(u, a_0^*, o_0^*)$ from $\mathcal{G}_{\text{drf}}$ to serve as the fixed seed state $s_1^*$. From this same state, we roll out $G$ refinement trajectories, obtaining the group of refinement trajectories:

$$\mathcal{G}_{\text{rfn}} = \left\{ \tau_{\text{rfn}}^{(i)} \right\}_{i=1}^{G}$$

$$= \left\{ \left( u, a_0^*, o_0^*, a_1^{(i)}, o_1^{(i)}, \cdots, a_T^{(i)}, o_T^{(i)} \right) \right\}_{i=1}^{G}. \tag{12}$$

Here, the actions are sampled sequentially as $a_t^{(i)} \sim \pi_{\theta_{\text{old}}} ( \cdot \mid s_t^{(i)} )$, with the fixed initial state $s_1^{(i)} = s_1^* = (u, a_0^*, o_0^*)$. Each trajectory in this group is evaluated using the refinement judge, yielding:

$$\mathbf{r}_{\text{rfn}} = \left\{ r_{\text{rfn}}^{(i)} \right\}_{i=1}^{G} = \left\{ \mathcal{R}_{\text{rfn}} \left( u, I_0^*, I_T^{(i)} \right) \right\}_{i=1}^{G}. \tag{13}$$

**Group-Normalized Advantages** We compute advantages separately within each group by normalizing rewards across the $G$ samples. For the draft group, the advantage for the $i$-th trajectory is computed as:

$$A_{\text{drf}}^{(i)} = \frac{r_{\text{drf}}^{(i)} - \text{mean}(\mathbf{r}_{\text{drf}})}{\text{std}(\mathbf{r}_{\text{drf}})} \tag{14}$$

We apply the same group-wise normalization to the refinement rewards $\mathbf{r}_{\text{rfn}}$ to obtain $A_{\text{rfn}}^{(i)}$ for each refinement trajectory.

### 3.2.4. POLICY OPTIMIZATION

We jointly optimize PosterAgent's drafting and refinement abilities by combining the two trajectory groups into a single training set:

$$\mathcal{G}_{\text{train}} = \mathcal{G}_{\text{drf}} \cup \mathcal{G}_{\text{rfn}}. \tag{15}$$

Each trajectory $\tau^{(i)} \in \mathcal{G}_{\text{train}}$ is associated with a stage-specific, group-normalized advantage $A^{(i)}$ computed in Sec. 3.2.3. Importantly, although the two stages are merged for a unified update, the time steps that participate in optimization depend on the trajectory source. Specifically, we define $\mathbf{T}^{(i)}$ as the set of time steps that participate in optimization (i.e., the steps whose action tokens are included in the training objective) for trajectory $\tau^{(i)}$:

$$\mathbf{T}^{(i)} = \begin{cases} \{0\}, & \tau^{(i)} \in \mathcal{G}_{\text{drf}} \\ \{1, \cdots, T\}, & \tau^{(i)} \in \mathcal{G}_{\text{rfn}}. \end{cases} \tag{16}$$

We optimize the policy $\pi_\theta$ using the following clipped objective with KL regularization:

$$\mathcal{J}(\theta) = \mathbb{E}_{u \sim \mathcal{U}, \mathcal{G}_{\text{train}} \sim \pi_{\theta_{\text{old}}} ( \cdot \mid u )}$$

$$\frac{1}{|\mathcal{G}_{\text{train}}|} \sum_{i=1}^{|\mathcal{G}_{\text{train}}|} \frac{1}{|\mathbf{T}^{(i)}|} \sum_{t \in \mathbf{T}^{(i)}} \frac{1}{\left| a_t^{(i)} \right|} \sum_{j=1}^{\left| a_t^{(i)} \right|} \left\{ - \beta \mathbb{D}_{\text{KL}} \right.$$

$$+ \min \left[ \rho_{t,j}^{(i)} A^{(i)}, \text{clip} \left( \rho_{t,j}^{(i)}, 1 - \epsilon, 1 + \epsilon \right) A^{(i)} \right] \bigg\}, \tag{17}$$

where $\rho_{t,j}^{(i)}$ is the token-level importance ratio for the $j$-th token of $a_t^{(i)}$:

$$\rho_{t,j}^{(i)} = \frac{\pi_\theta \left( a_{t,j}^{(i)} \middle| s_t^{(i)}, a_{t,1:j-1}^{(i)} \right)}{\pi_{\theta_{\text{old}}} \left( a_{t,j}^{(i)} \middle| s_t^{(i)}, a_{t,1:j-1}^{(i)} \right)}. \tag{18}$$

## 4. Experiment

### 4.1. Data Preparation

We construct the instruction corpus for both training and evaluation via an automated LLM-driven pipeline, leveraging state-of-the-art proprietary models (e.g., the GPT and Gemini families) as generators. Our data collection pipeline follows a hierarchical strategy designed to maximize coverage across eight distinct domains: Technology & Innovation, Science & Discovery, Health & Wellness, History & Culture, Education & Learning, Art & Creativity, Business & Finance, and Lifestyle & Hobbies. The generation process consists of two stages:

1. **Meta-Instruction Generation:** For each domain, the generator model is prompted to produce a set of meta-instructions: short, high-level briefs that specify only the essential poster intent (i.e., "what the poster is about") while deliberately leaving stylistic and layout decisions under-specified.

2. **Detailed Expansion:** To simulate the complexity of real-world design tasks, we expanded each meta-instruction into multiple distinct user instructions. While sharing

the same core topic, these derived instructions vary significantly in their constraints—specifying unique combinations of target audience, communication goals, information hierarchy, layout constraints, visual tone, and stylistic preferences.

All meta-instructions and their expanded user instructions underwent rigorous human verification to filter out ambiguous or low-quality samples. The resulting dataset contains 1,000 user instructions per domain (8,000 total). We hold out 200 instructions for testing and use the remaining 7,800 for training. Crucially, to rigorously assess generalization, we enforced a strict separation at the meta-instruction level. This ensures that the test set contains no user instructions derived from meta-instructions present in the training set, thereby preventing the model from succeeding via memorization of specific topic patterns.

To facilitate policy initialization, we randomly sample approximately 3,000 training instructions and use Gemini 2.5 Pro to synthesize expert trajectories in our interactive design format. These trajectories are used for cold-start training. The remaining instructions are reserved for the proposed Stage-Aware Reinforcement Learning phase.

### 4.2. Experimental Setup

**Benchmarks** We evaluate our method on two benchmarks. (1) **PosterAgent-Test** is an in-domain evaluation set consisting of 200 held-out user instructions from our instruction corpus; the split is performed at the meta-instruction level to prevent topic leakage between train and test (Sec. 4.1). (2) **Out-of-Domain Benchmark** is an external public graphic-design benchmark introduced in prior work (Jia et al., 2023). It contains professional graphic-design intention prompts across diverse design scenarios. Following the evaluation protocol of the original work, we use GPT-4V as the evaluator and score each generated design on five dimensions: design and layout, content relevance and effectiveness, typography and color scheme, graphics and images, innovation and originality. Notably, this benchmark differs substantially from our training distribution in artifact types, prompt style, and evaluation metrics, and is therefore used to assess out-of-domain generalization.

**Baselines** We comprehensively compare PosterAgent against strong baselines, including: (1) Text-to-image generation models: we directly prompt several strong contemporary image generators, including Qwen-Image (Wu et al., 2025a), HunyuanImage 3.0 (Cao et al., 2025), Seedream 3.0 (Gao et al., 2025), and Seedream 4.0 (Seedream et al., 2025), to produce the final design in an end-to-end manner. (2) Frontier LLMs: we evaluate both closed-source and open-weight frontier LLMs, including GPT-5 (Singh et al., 2025), Gemini 2.5 Pro (Comanici et al., 2025), DeepSeek-

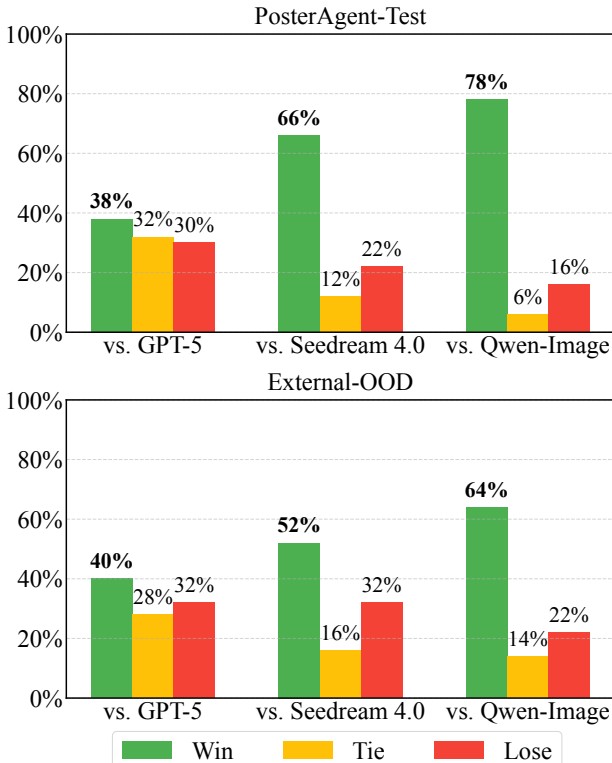

*Figure 2.* Human evaluation results on PosterAgent-Test and the external out-of-domain benchmark.

V3.2 (Liu et al., 2025), and Kimi K2 (Team et al., 2025), in a zero-shot setting, where the model is given our design schema and asked to output a complete structured design that can be rendered into the final poster. (3) Structured intermediate-representation pipelines: on the external out-of-domain benchmark, we additionally compare with prior graphic-design generation approaches that explicitly predict layered, editable design elements, including COLE (Jia et al., 2023) and OpenCOLE (Inoue et al., 2024).

**Implementation Details** To balance generation quality with computational efficiency, we instantiate PosterAgent with Qwen3-VL-8B as the backbone model. Training proceeds in two stages. First, we perform a cold-start supervised fine-tuning on collected expert trajectories, which teaches the model to follow our predefined design schema and equips it with basic multi-turn interaction skills. Second, we apply reinforcement learning to further improve decision-making and overall design performance beyond the supervised initialization. We set the number of refinement steps to $T = 3$. Full training configurations and hyperparameters are provided in Appendix C.

*Table 1.* Performance comparison of different methods on our PosterAgent-Test benchmark. Each generated poster is evaluated by GPT-5 based on the following criteria: text quality, image quality, layout and typography, instruction adherence and overall aesthetics. The best results among open-source methods are in **bold**, and the best results among closed-source methods are underlined.

| Methods | Open-Source | Text | Image | Layout | Instruction | Aesthetics | Overall |
|---------|:-----------:|:----:|:-----:|:------:|:-----------:|:----------:|:-------:|
| *Text-to-image Models* | | | | | | | |
| HunyuanImage 3.0 | ✓ | 1.88 | 6.94 | 6.61 | 3.50 | 4.38 | 4.66 |
| Qwen-Image | ✓ | 2.29 | 6.52 | 6.18 | 4.10 | 4.50 | 4.72 |
| Seedream 3.0 | ✗ | 2.23 | 6.96 | 6.57 | 4.10 | 4.60 | 4.89 |
| Seedream 4.0 | ✗ | 5.04 | 7.24 | 6.62 | 6.34 | 6.15 | 6.28 |
| *Frontier (M)LLMs* | | | | | | | |
| DeepSeek-V3.2 | ✓ | 5.14 | 5.72 | 5.14 | 4.74 | 4.95 | 5.14 |
| Kimi K2 | ✓ | 6.52 | 7.09 | 6.39 | 6.16 | 6.24 | 6.48 |
| Gemini 2.5 Pro | ✗ | 7.38 | 7.46 | 7.19 | 7.32 | 7.20 | 7.31 |
| GPT-5 | ✗ | 7.34 | 7.58 | 7.14 | 7.80 | 7.30 | 7.43 |
| Qwen3-VL-8B | ✓ | 5.06 | 6.04 | 4.74 | 4.77 | 4.76 | 5.08 |
| PosterAgent (Cold-Start) | ✓ | 6.74 | 7.11 | 6.70 | 6.64 | 6.64 | 6.76 |
| **PosterAgent** | ✓ | **7.32** | **7.60** | **7.44** | **7.38** | **7.29** | **7.41** |

## 4.3. Main Results

We present the quantitative evaluation results on the in-domain PosterAgent-Test (Table 1) and the out-of-domain benchmark (Table 2). Our analysis yields three primary conclusions regarding the efficacy of the PosterAgent framework.

**Superior Text Rendering and Content Planning over T2I Models**   A key advantage of PosterAgent over text-to-image (T2I) baselines lies in its ability to produce both visually legible text and semantically appropriate, instruction-aligned content. In our evaluation, the "Text Quality" metric measures not only readability but also the coherence and appropriateness of the textual content relative to user intent. As shown in Table 1, T2I models struggle with this dual requirement: HunyuanImage 3.0 and Qwen-Image obtain particularly low text scores (1.88 and 2.29), and even the strongest T2I baseline Seedream 4.0 reaches only 5.04. In contrast, PosterAgent achieves a substantially higher Text Quality score of 7.32. These results demonstrate that combining an editable design specification with iterative refinement significantly improves text quality for poster generation.

**Competitive Performance with Frontier Proprietary Models.**   Despite utilizing an 8B-parameter open-weight backbone, PosterAgent demonstrates performance parity with closed-source frontier models. On the PosterAgent-Test (Table 1), our method surpasses Gemini 2.5 Pro and is statistically comparable to GPT-5. Crucially, this competitiveness also holds in the out-of-domain setting (Table 2): PosterAgent achieves the best overall score of 7.66, slightly exceeding GPT-5 and clearly outperforming Gemini 2.5 Pro. This result suggests that a smaller, specialized agent equipped with a rigorous self-refinement loop can match

the reasoning and design capabilities of significantly larger general-purpose models.

**Generalization to Out-of-Domain Scenarios.**   Results on the external out-of-domain benchmark demonstrate strong generalization beyond our training distribution in both prompt style and evaluation criteria. PosterAgent achieves a state-of-the-art overall score of 7.66, outperforming all baselines. Notably, PosterAgent significantly surpasses existing structured layout generation methods such as OpenCOLE and COLE. Unlike these traditional approaches that rely on one-shot static predictions, our agent's iterative refinement capability allows it to adaptively improve typography and composition, yielding superior results even on design tasks with different distributions and evaluation criteria.

## 4.4. Human Evaluation

To complement the automatic MLLM-based evaluation, we conduct a human preference study on both the PosterAgent-Test set and the external out-of-domain benchmark. Specifically, we randomly sample 50 user instructions from each evaluation dataset. For every instruction, we present annotators with the instruction and two anonymized posters: one generated by PosterAgent and the other by a baseline, and ask them to determine which one is better. Results are shown in Figure 2.

As illustrated, PosterAgent is highly competitive with GPT-5, achieving slightly higher win rates on both benchmarks. Against T2I models, PosterAgent demonstrates clear superiority, particularly on PosterAgent-Test, winning 66% against Seedream 4.0 and 78% against Qwen-Image. Notably, the margin over T2I models slightly narrows on the external benchmark. This is primarily because the external

*Table 2.* Performance comparison of different methods on the external out-of-domain graphic-design benchmark. Following the evaluation protocol of prior work, we employ GPT-4V to assess the posters based on the following dimensions: design and layout, content relevance and effectiveness, typography and color scheme, graphics and images, innovation and originality. The best results among open-source methods are in **bold**, and the best results among closed-source methods are underlined.

| Methods | Open-Source | Design | Content | Typography | Graphic | Innovation | Overall |
|---|---|---|---|---|---|---|---|
| *Text-to-image Models* | | | | | | | |
| HunyuanImage 3.0 | ✓ | 7.67 | 6.74 | 6.54 | **8.60** | **6.36** | 7.18 |
| Qwen-Image | ✓ | 7.26 | 6.80 | 6.38 | 8.10 | 5.89 | 6.88 |
| Seedream 3.0 | ✗ | 7.64 | 6.66 | 6.83 | 8.57 | 6.10 | 7.16 |
| Seedream 4.0 | ✗ | 7.80 | 7.68 | 7.43 | 8.39 | 6.04 | 7.47 |
| *Frontier (M)LLMs* | | | | | | | |
| DeepSeek-V3.2 | ✓ | 6.24 | 6.70 | 5.92 | 7.13 | 5.11 | 6.22 |
| Kimi K2 | ✓ | 6.65 | 7.26 | 6.32 | 7.36 | 5.47 | 6.61 |
| Gemini 2.5 Pro | ✗ | 7.49 | 7.72 | 7.30 | 7.92 | 5.84 | 7.25 |
| GPT-5 | ✗ | 7.85 | 8.10 | 7.58 | 8.43 | 6.20 | 7.63 |
| *Structured IR-based Methods* | | | | | | | |
| COLE | ✗ | 6.00 | 6.90 | 5.70 | 6.20 | 5.10 | 6.00 |
| OpenCOLE | ✓ | 6.30 | 7.00 | 5.60 | 7.10 | 5.30 | 6.30 |
| Qwen3-VL-8B | ✓ | 6.18 | 6.82 | 5.98 | 7.21 | 5.63 | 6.36 |
| PosterAgent (Cold-Start) | ✓ | 6.95 | 7.30 | 6.75 | 7.63 | 5.72 | 6.87 |
| PosterAgent | ✓ | **7.81** | **8.18** | **7.60** | 8.50 | 6.20 | **7.66** |

benchmark prompts are generally simpler, involving less textual content and more straightforward layouts. Nonetheless, PosterAgent maintains consistent advantages across both settings, validating the efficacy of the agentic design paradigm.

### 4.5. Ablation Studies

We conduct ablation studies on PosterAgent-Test to isolate the contributions of key components in our framework. All ablation variants were re-trained from scratch, including the cold-start phase, to ensure a fair comparison.

**Effectiveness of the Iterative Agentic Workflow.** We first examine the importance of the agentic paradigm by comparing against a single-turn variant (w/o self-refinement), where the model generates the final poster layout in one shot without any iterative observation and modification. As shown in Table 3, removing the multi-turn refinement capability causes a significant performance drop, with the overall score decreasing from 7.41 to 6.80. This validates that the iterative refinement process is crucial for high-quality poster generation.

**Effect of Stage-Aware Reward Design.** To verify the effectiveness of our SARL framework, we compare against a variant (w/o stage-aware reward) that employs a conventional outcome-only reward scheme. In this ablation, although the model still performs multi-turn refinement during inference, the RL training assigns a single reward based

solely on the final poster quality to all actions in the trajectory. The results demonstrate a notable degradation in performance (overall score drops from 7.41 to 6.95), confirming that conflating drafting and refinement signals leads to noisy credit assignment. See Appendix B for further analysis of the reward design.

**Effect of Diagnostic Feedback.** We also investigate the impact of textual feedback from the rendering environment by removing the execution warnings (w/o warnings). These warnings provide symbolic feedback regarding text overflow, invalid geometric parameters, and other layout inconsistencies that are difficult to detect purely from pixel-level visual observation. Without this diagnostic information, the model's overall score decreases to 7.16. This indicates that the structured textual observations complement visual feedback by enabling targeted fixes for specific functional constraints, thereby improving both typography quality and layout validity.

**Impact of Refinement Steps.** Figure 3 illustrates how performance scales with the number of refinement iterations $T$. When $T = 0$ (no refinement, equivalent to the w/o self-refinement ablation), the model achieves an overall score of 6.80. The performance improves progressively as $T$ increases from 0 to 3, reaching a peak of 7.41 at $T = 3$. Beyond this point, performance plateaus when $T$ is increased to 4, suggesting that while iterative refinement consistently enhances quality, the marginal gains diminish after three iterations. This saturation effect likely occurs because most

*Table 3.* Results of ablation studies on our PosterAgent-Test benchmark.

| Methods | Text | Image | Layout | Instruction | Aesthetics | Overall |
|---|---|---|---|---|---|---|
| Qwen3-VL-8B | 5.06 | 6.04 | 4.74 | 4.77 | 4.76 | 5.08 |
| PosterAgent | 7.32 | 7.60 | 7.44 | 7.38 | 7.29 | 7.41 |
| w/o self-refinement | 6.72 | 7.08 | 6.78 | 6.70 | 6.74 | 6.80 |
| w/o stage-aware reward | 6.90 | 7.18 | 6.84 | 6.96 | 6.88 | 6.95 |
| w/o warnings | 7.04 | 7.32 | 7.20 | 7.17 | 7.07 | 7.16 |

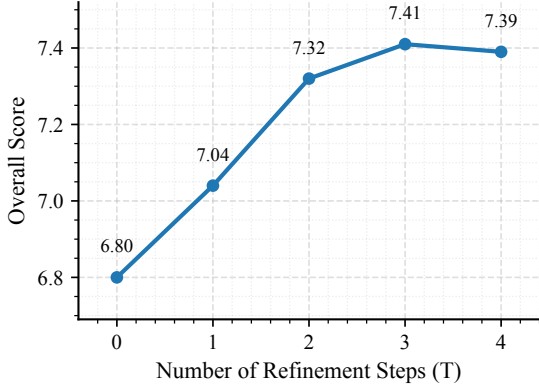

*Figure 3.* Performance on PosterAgent-Test under different numbers of refinement steps.

correctable layout and typography issues are resolved within the first few refinement rounds, and further edits risk introducing minor unnecessary variations or over-optimization on negligible details.

## 5. Conclusion

In this work, we introduce PosterAgent, a novel agentic framework that reframes automated poster generation as an iterative workflow. By modeling the design process as a POMDP and employing a dedicated Stage-Aware Reinforcement Learning paradigm, we decouple the learning signals for drafting and refinement, enabling more precise credit assignment and stable policy optimization. Extensive experiments demonstrate that PosterAgent achieves competitive performance with state-of-the-art proprietary models, excels in text rendering and layout planning, and generalizes robustly to out-of-domain scenarios. Our work illustrates how a specialized, modular agent equipped with structured reasoning and self-critique can effectively automate complex design workflows, paving the way for more adaptive and controllable generative systems.

## Limitations

Our study has several limitations that suggest directions for future work. First, our reward relies on an MLLM-as-Judge scheme, which, while scalable, may still diverge from professional designers on subtle aesthetic dimensions; a natural extension is to curate human-annotated preference data and train a specialized aesthetic reward model. Second, we instantiate our framework only on poster generation, but its core formulation, which casts structured design as an iterative POMDP with decoupled, stage-aware rewards, is largely domain-agnostic and should transfer naturally to broader design tasks such as web design, UI layout, and slide creation. Finally, given the rapid progress of text-to-image models toward highly realistic imagery, a compelling direction is to more tightly integrate structured, editable design representations with state-of-the-art generative models, combining the controllability of layout-driven approaches with the visual fidelity of modern T2I systems.

## Impact Statement

This paper presents work whose goal is to advance the field of Machine Learning. There are many potential societal consequences of our work, none of which we feel must be specifically highlighted here.

## Acknowledgements

We thank anonymous reviewers for their helpful comments on this paper. This work was partially supported by the National Natural Science Foundation of China (Grant No. 62476010).

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

# A. Case Study

## A.1. Comparison with Other Methods

We include two case studies comparing PosterAgent with other methods in Figures 5 and 6. Qualitative analysis of the generated posters demonstrates the distinct advantages of PosterAgent in design quality and instruction adherence. While Qwen-Image fundamentally struggles to render legible and correct text, Seedream 4.0 shows relative improvement in visual synthesis but still frequently renders erroneous characters and fails to organize content logically according to complex user instructions. Even frontier models such as GPT-5, despite their strong general capabilities, lack an intrinsic self-reflection mechanism to fix rendering flaws, often resulting in visible inconsistencies such as uneven font sizes. In contrast, PosterAgent effectively leverages its iterative self-refinement loop to identify and resolve layout issues, consistently producing professional designs with superior text rendering and structural harmony.

## A.2. Effectiveness of the Iterative Self-Refinement Paradigm

Figure 7 illustrates the progressive evolution of a poster through the iterative self-refinement paradigm of PosterAgent. The initial draft (leftmost) exhibits a chaotic layout with overlapping elements and poor information hierarchy, which significantly hinders readability. Through three successive rounds of critique and modification, the framework systematically addresses these structural flaws—refining alignment, optimizing spatial distribution, and clarifying the visual flow. The final result (rightmost) demonstrates a well-organized and professional layout, confirming that the multi-turn agentic process is essential for transforming a cluttered conceptual draft into a high-quality, aesthetically pleasing design.

# B. Analysis of Stage-Aware Rewards

In this section, we analyze the impact of different reward mechanisms on the agent's actual engagement in the refinement process. In our work, although the refinement horizon is fixed at $T = 3$ for all experiments, we observe that the reward structure significantly influences whether the agent utilizes these turns effectively. We define a valid refinement step as an iteration where the agent produces a modified JSON design specification that differs from the previous turn, indicating a purposeful attempt to improve the design.

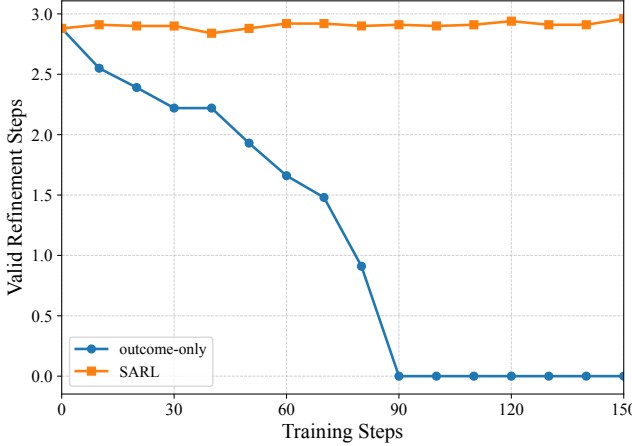

*Figure 4.* Trends in the number of valid refinement steps under different reward settings. We define a valid refinement step as one in which the JSON is modified (making it different from the previous step).

As illustrated in Figure 4, the conventional outcome-only reward setting leads to a rapid degradation of refinement behavior during training. While the agent initially attempts to refine designs, the number of valid refinement steps progressively declines and collapses to zero after approximately 90 training steps. This suggests that under an outcome-only signal, the agent learns to "bypass" the refinement stage by outputting identical JSON specifications, likely because the noisy credit assignment fails to incentivize the marginal value of editing beyond the quality of the initial draft. In contrast, PosterAgent trained with SARL maintains a consistently high number of valid refinement steps throughout the entire training process, remaining stable near the maximum horizon of $T = 3$. By decoupling the reinforcement signals, SARL provides a clear, stage-specific advantage for effective modifications through the relative improvement judge. This confirms that our

framework effectively prevents the collapse of agentic behavior and ensures that the multi-turn capability is fully utilized to optimize the final design.

## C. Implementation Details

**Renderer**    The deterministic renderer $\mathcal{M}$ described in Sec. 3.1.2 is built on Skia, an open-source 2D graphics library. For image elements, the agent provides a prompt and a bounding box (bbox_2d) in JSON. The renderer then invokes Seedream 4.0 (Seedream et al., 2025) to generate images matching the specified dimensions. It also performs layout sanity checks to return symbolic execution warnings ($W_t$).

**MLLM Judge**    We employ GPT-5 as the backbone for both the Draft Judge and Refinement Judge. The former scores initial drafts on a 1–9 scale across five design dimensions, while the latter evaluates relative improvement between iterations on a 0–4 scale (see Appendix D.2 for more details).

**Training Infrastructure**    The cold-start training is based on the LLaMA-Factory (Zheng et al., 2024) framework, and the reinforcement learning training is based on the verl framework. Detailed hyperparameter settings for both the cold-start and SARL phases are provided in Tables 4 and 5, respectively.

*Table 4.* Hyperparameter settings for the cold-start phase.

| Hyperparameter | Value |
| --- | --- |
| Learning Rate | 1e-5 |
| Batch Size | 128 |
| Number of Epochs | 3 |
| Warmup Ratio | 0.05 |
| LR Scheduler Type | Cosine with Min LR |
| Max Context Length | 32768 |

*Table 5.* Hyperparameter settings for the SARL phase.

| Hyperparameter | Value |
| --- | --- |
| Learning Rate | 1e-6 |
| Batch Size | 32 |
| Group Size ($G$) | 8 |
| Temperature | 1.0 |
| Top-p | 0.95 |
| KL Coefficient ($\beta$) | 0 |
| Number of Epochs | 1 |
| Max Context Length | 32768 |
| Number of Refinement Steps ($T$) | 3 |

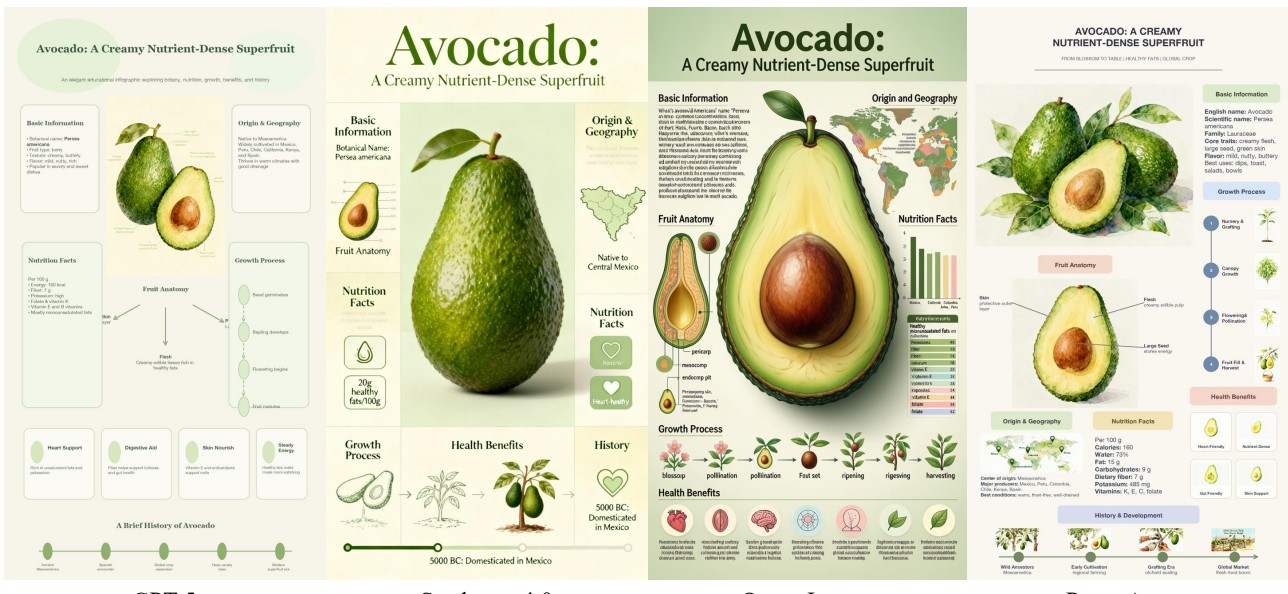

GPT-5       Seedream 4.0       Qwen-Image       PosterAgent

*Figure 5.* Case study with **user instruction:** Create an elegant educational infographic titled "Avocado: A Creamy Nutrient-Dense Superfruit." Use soft green, cream, and pastel tones to give the poster a fresh, natural, and polished appearance. Feature a large avocado illustration as the central visual, then organize the content into multiple parallel sections such as basic information, fruit anatomy, origin and geography, nutrition facts, growth process, health benefits, and history. Use a multi-column layout where different content streams can run side by side — for example, pairing a visual diagram on one side with a step-by-step process on the other. Include small supporting visuals like labeled diagrams, icon-style benefit boxes, and a horizontal timeline at the bottom. The layout should feel balanced and information-rich without being crowded. The overall atmosphere should be refined, botanical, and visually soothing.

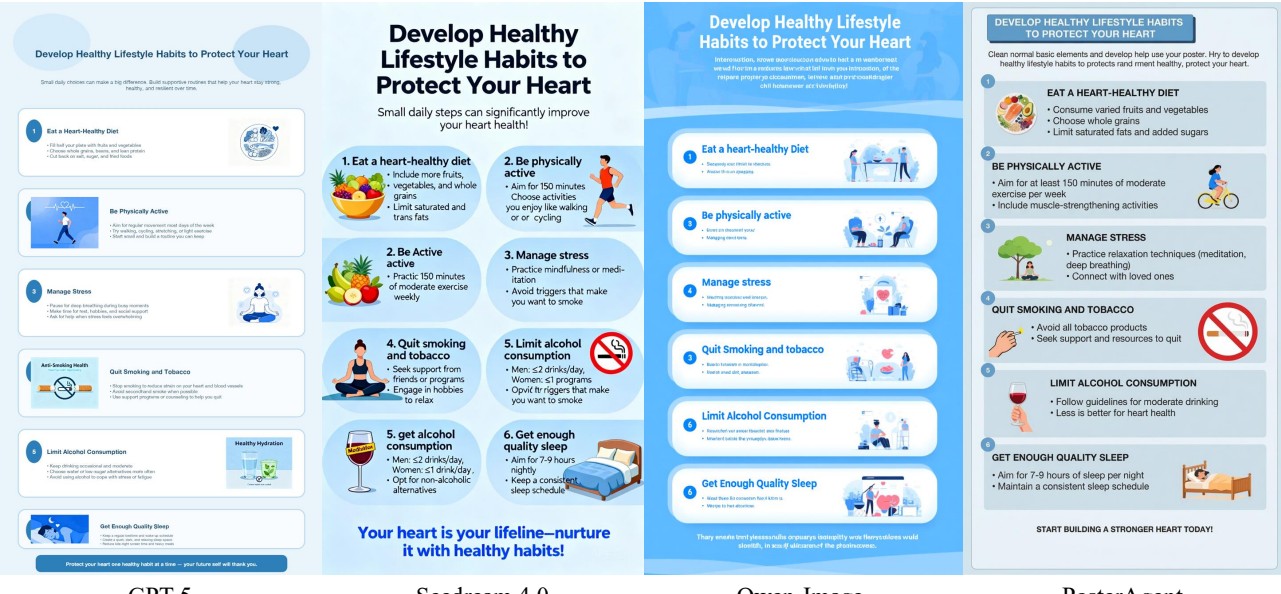

GPT-5       Seedream 4.0       Qwen-Image       PosterAgent

*Figure 6.* Case study with **user instruction:** Design a clean and encouraging health poster titled "Develop Healthy Lifestyle Habits to Protect Your Heart." Use a calm blue color palette, rounded content boxes, and simple health-related illustrations to create a professional and reassuring look. Add a short introduction under the title, then organize the poster into six clearly numbered sections: eat a heart-healthy diet, be physically active, manage stress, quit smoking and tobacco, limit alcohol consumption, and get enough quality sleep. Each section should include a bold heading, a few practical bullet points, and a matching visual. Vary the placement of illustrations across sections — sometimes on the left, sometimes on the right — to create visual rhythm and avoid a monotonous layout. End with a strong motivational message at the bottom. The overall design should feel clear, positive, and easy to follow.

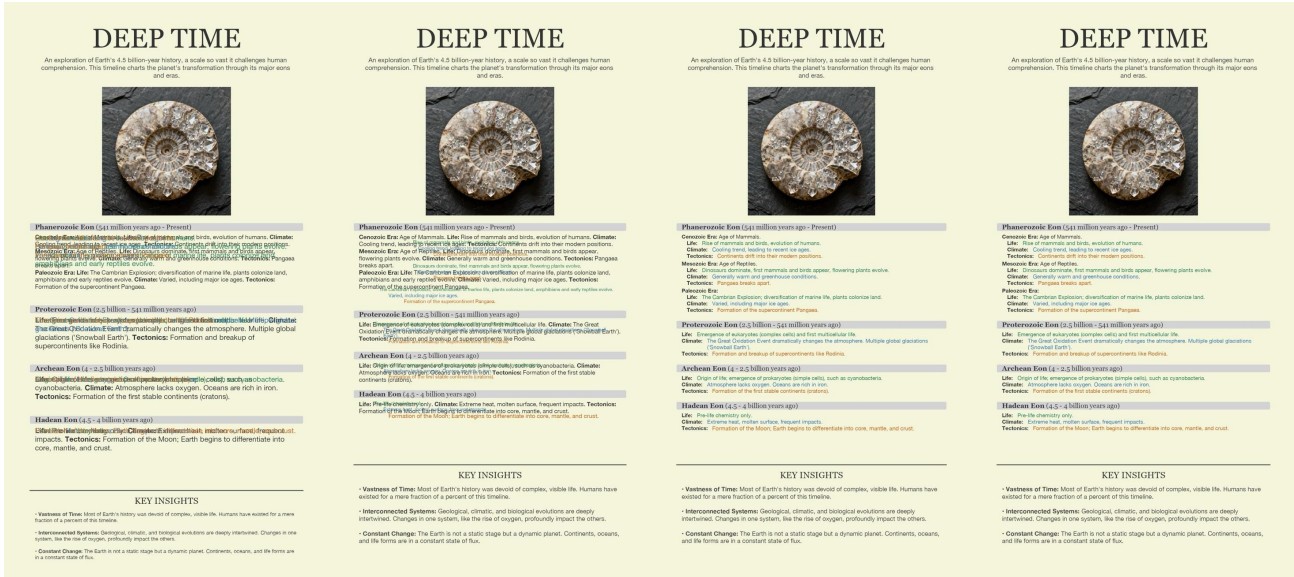

*Figure 7.* Evolution of poster generation through iterative refinement. The process begins with a cluttered initial draft that lacks clear hierarchy, which is then systematically transformed into a professionally organized and readable design over three refinement turns.

## D. Prompts

### D.1. Prompt of PosterAgent

---

**Prompt of PosterAgent**

You are an expert AI Poster Designer. Your mission is to create a professional, beautifully designed vertical poster based on user requests by iteratively using a "Render" tool.

**Your Workflow (Mandatory)**
You must follow this iterative "Design-Render-Reflect" process:
1. **Plan the Layout:** First, carefully analyze the user's request and consider the optimal design for the poster.
2. **Call the Render Tool and STOP:** Create a **complete design** as a single JSON object and call the "Render" tool. Every tool call must contain all elements and their attributes for the poster. You must then STOP and WAIT for the user to provide the tool's return.
3. **Review the Output and Warnings:** The tool will return the rendered poster and a list of warnings. You must read every warning message and critically examine every element you designed as well as the overall visual effect, and try your best to identify areas that can still be improved.
4. **Modify the Design:** You should first think about how to modify your design to fix the issues and make it better. You can add or remove elements, or adjust their properties. Then call "Render" again with **a new, complete JSON object** as the argument.

**Tool Guide: The "Render" Tool**
**1. Call Format**
You must strictly follow this format, wrapping the JSON object in <tool_call> tags:
<tool_call>
{"name": "Render", "arguments": {"design_elements": [{ "element_type": "text", ... }, { "element_type": "image", ... }]}}
</tool_call>

**2. Design Elements**

---

The "design_elements" key must contain a non-empty list of element objects. There are five element types:

**A. Text Element**

Description: This represents a text box on the canvas. The tool will render the text provided in the "content" property within the boundaries defined by "bbox_2d".

Attributes:

- element_type: "text"
- bbox_2d: [x1, y1, x2, y2] (relative integer coordinates in [0,1000]; [x1,y1] is top-left, [x2,y2] is bottom-right)
- content: Your text with **bold** support. (string; English only; supports using double asterisks for bolding)
- font_family: "Helvetica Neue", "Georgia" or "Comic Sans MS" (string)
- font_size: 30 (integer)
- color: "#RRGGBB", "#RRGGBBAA" or "color_name" (string; supports hex codes and common color names including "black", "white", "red", "green", "blue", "yellow", "cyan", "magenta", "gray", "orange" and "purple")
- horizontal_align: "left", "center" or "right" (string)
- vertical_align: "top", "middle" or "bottom" (string)

**B. Image Element**

Description: This represents an image frame on the canvas. The tool will first generate an image based on the "prompt" and the size specified by "bbox_2d", and then place the generated image into this frame.

Attributes:

- element_type: "image"
- bbox_2d: [x1, y1, x2, y2] (relative integer coordinates in [0,1000]; [x1,y1] is top-left, [x2,y2] is bottom-right)
- prompt: "A description used to generate the image." (string; do not specify image size here)

**C. Line (Arrow) Element**

Description: This represents a straight line. It can be solid or dashed, and can optionally have an arrowhead at the endpoint.

Attributes:

- element_type: "line"
- p1: [x1, y1] (relative integer coordinates in [0,1000] for the start point)
- p2: [x2, y2] (relative integer coordinates in [0,1000] for the endpoint)
- width: 10 (integer)
- color: "#RRGGBB", "#RRGGBBAA", or "color_name" (string)
- is_dashed: false (boolean; set to true for a dashed line)
- has_arrow: false (boolean; set to true to add an arrowhead at p2)

**D. Rect Element**

Description: This represents a rectangle, which can have rounded corners.

Attributes:

- element_type: "rect"
- bbox_2d: [x1, y1, x2, y2] (relative integer coordinates in [0,1000]; [x1,y1] is top-left, [x2,y2] is bottom-right)
- fill_color: "#RRGGBB", "#RRGGBBAA", "color_name" or "transparent" (string)
- stroke_width: 5 (integer; set to 0 for no outline)
- stroke_color: "#RRGGBB", "#RRGGBBAA" or "color_name" (string; ignored if "stroke_width" is 0)
- corner_radius: 15 (integer)

**E. Ellipse Element**

Description: This represents an ellipse or a circle, inscribed within the given bounding box.

Attributes:

- element_type: "ellipse"
- bbox_2d: [x1, y1, x2, y2] (relative integer coordinates in [0,1000]; [x1,y1] is top-left, [x2,y2] is bottom-right)
- fill_color: "#RRGGBB", "#RRGGBBAA", "color_name" or "transparent" (string)
- stroke_width: 5 (integer; set to 0 for no outline)

• stroke_color: "#RRGGBB", "#RRGGBBAA" or "color_name" (string; ignored if "stroke_width" is 0)

**Understanding the Tool's Response**
The tool's return value contains the rendered poster and any warnings.
**1. Response Format**
<tool_response>
[Warning] ... (zero or more lines, each warning applies to one element index)
Here is the rendered poster: <rendered_poster>
</tool_response>

**2. How to Handle Warnings**
Each warning applies to one element index. Indexes start at zero.

**Type 1: Element Render Failure Warning**
What it means: The tool could not render an element, likely due to a malformed "bbox_2d" or other invalid parameters.
Your Action: You **must** correct the parameters for that element in your next design attempt.

**Type 2: Font Size Reduction Warning**
What it means: The "font_size" you chose was too large for the text "content" to fit in the specified "bbox_2d". To prevent overflow, the tool automatically shrank the font.
Your Action (Choose one of three options):
A. Accept the Smaller Font: In your new JSON, update the element with the new, smaller "font_size". **For visual consistency, you should also consider reducing the font size of other similar elements.**
B. Enlarge the Text Box: Keep your original "font_size" but increase the size of the "bbox_2d" in your new JSON.
C. Shorten the Text: Keep your original "font_size" and "bbox_2d" but edit the "content" to be shorter in your new JSON.

**Important Notes**
1. Your ultimate goal is to create a poster that is well-composed, visually appealing, and fulfills the user's request. Eliminating all warnings is a means to achieve this goal, but **not the goal itself**. You should critically examine every element you designed as well as the overall visual effect of each iteration, and try your best to identify areas that can still be improved. The final quality of the poster and its adherence to the user's requirements should always be your top priority.
2. Before you call the "Render" tool, you MUST explicitly output your thought process.
3. To avoid an excessively long context, only the most recent two designs and their corresponding posters will be carried over. JSON designs and rendered posters from earlier iterations will be omitted in subsequent turns.

## D.2. Prompt of MLLM Judge

**Prompt of Draft Judge**

You are a **STRICT** AI design critic. Your task is to evaluate a poster that was automatically generated based on a user's text instruction. You will be given the user's instruction and the generated poster image.
You must evaluate the poster across five dimensions: **Text Quality**, **Image Quality**, **Layout & Composition**, **Instruction Adherence**, and **Overall Aesthetics & Impact**. For each dimension, you will assign a score from 1 to 9.
Your final output must be a single, valid JSON object containing the scores for the five dimensions.

**Evaluation Criteria and Scoring Rubric**
**1. Text Quality:** Evaluates the text's correctness, legibility, and, most importantly, its **appropriateness for the poster's theme and purpose**.

- 9 (Excellent): The text is perfectly legible with no errors. Crucially, its content and tone are highly compelling and perfectly aligned with the poster's theme and the user's intent.
- 8: Performance between "Excellent" and "Good".
- 7 (Good): The text is clear with only minor, if any, errors. The content is relevant and effectively supports the poster's message and instruction.
- 6: Performance between "Good" and "Acceptable".
- 5 (Acceptable): The text is generally legible, though it may have some errors. The content is relevant to the topic, but may be generic or not fully capture the desired tone.
- 4: Performance between "Acceptable" and "Poor".
- 3 (Poor): The text contains significant errors hindering readability. Alternatively, the content itself is a poor fit for the poster's theme, even if grammatically correct.
- 2: Performance between "Poor" and "Unacceptable".
- 1 (Unacceptable): The text is illegible, contains critical errors, or its content is completely irrelevant and detrimental to the poster's purpose.

**2. Image Quality:** Evaluates the image's technical quality and, most importantly, its **relevance and contribution to the poster's message and requested style**.
- 9 (Excellent): The image is high-resolution, sharp, and free of distortions. Critically, the visuals are highly relevant, powerfully enhance the poster's message, and perfectly match the style requested in the instruction.
- 8: Performance between "Excellent" and "Good".
- 7 (Good): The image is high-resolution with minimal visual inconsistencies. The visuals are relevant, stylistically appropriate for the theme, and support the poster's purpose.
- 6: Performance between "Good" and "Acceptable".
- 5 (Acceptable): The image has sufficient resolution, but may have noticeable artifacts. Thematically, the visuals are relevant but might not perfectly match the desired style or might fail to strongly enhance the poster's message.
- 4: Performance between "Acceptable" and "Poor".
- 3 (Poor): The image is low-resolution, blurry, or has distracting artifacts. Alternatively, the visuals are technically acceptable but are irrelevant or clash with the poster's theme.
- 2: Performance between "Poor" and "Unacceptable".
- 1 (Unacceptable): The image is corrupted or unrecognizable. Or, its content is completely irrelevant and undermines the purpose of the poster.

**3. Layout & Composition:** The arrangement and integration of all visual elements (text, images, shapes).
- 9 (Excellent): The layout is professionally balanced, with a clear visual hierarchy that effectively guides the viewer's eye. Excellent use of negative space (which serves to frame the content rather than creating empty voids), alignment, and typography. All elements are perfectly integrated.
- 8: Performance between "Excellent" and "Good".
- 7 (Good): The layout is well-organized and visually balanced. Hierarchy is clear, and text is easy to read. The composition is effective, with appropriate content density, but could be slightly improved.
- 6: Performance between "Good" and "Acceptable".
- 5 (Acceptable): The layout is functional but basic. Elements are placed without major clashes, but there is little to no clear hierarchy or design principle applied. The layout may feel slightly empty or loosely structured. Legibility is acceptable.
- 4: Performance between "Acceptable" and "Poor".
- 3 (Poor): The layout is either cluttered, unbalanced, or excessively sparse and empty. Elements overlap improperly, or large gaps make the poster feel unfinished. There is no clear focal point.
- 2: Performance between "Poor" and "Unacceptable".
- 1 (Unacceptable): The layout makes the poster incomprehensible. Key information is obscured, text is unreadable, or the arrangement of elements is completely nonsensical.

**4. Instruction Adherence:** How well the poster fulfills the user's original instruction.
- 9 (Excellent): The poster perfectly and creatively captures all aspects of the user's instruction, including theme,

subject matter, style, specific text, and any other explicit requests. It fully realizes the user's intent.
- 8: Performance between "Excellent" and "Good".
- 7 (Good): The poster addresses all major components of the instruction but may miss or misinterpret a minor detail. The overall result is very close to the user's request.
- 6: Performance between "Good" and "Acceptable".
- 5 (Acceptable): The poster captures the main idea of the instruction but ignores several specific details or includes unrequested elements. The core request is met, but the execution is imprecise.
- 4: Performance between "Acceptable" and "Poor".
- 3 (Poor): The poster only vaguely relates to the instruction. It ignores key constraints and fails to deliver on the core theme or content requested by the user.
- 2: Performance between "Poor" and "Unacceptable".
- 1 (Unacceptable): The poster completely disregards or contradicts the user's instruction.

**5. Overall Aesthetics & Impact:** Evaluates the overall aesthetic appeal, professionalism, and the synergistic effect of all elements combined.
- 9 (Excellent): The poster as a whole exhibits a high degree of professionalism and a unified visual style. All elements merge seamlessly to create a strong visual impact, perfectly conveying the core message and atmosphere.
- 8: Performance between "Excellent" and "Good".
- 7 (Good): The poster looks harmonious and professional overall. The elements work well together, and the visual effect is appealing, but there might be slight room for improvement in visual impact or stylistic consistency.
- 6: Performance between "Good" and "Acceptable".
- 5 (Acceptable): The poster is functionally complete but lacks a professional design touch. The elements are combined without obvious clashes but fail to produce a standout synergy, resulting in a mediocre overall impression.
- 4: Performance between "Acceptable" and "Poor".
- 3 (Poor): The poster's overall style is inconsistent, with noticeable clashes between elements (e.g., mismatched color tones, styles). The overall feel is chaotic, amateurish, and fails to establish an effective focal point.
- 2: Performance between "Poor" and "Unacceptable".
- 1 (Unacceptable): The poster is a completely disorganized collage of elements that severely clash, lacking any sense of design and failing to convey a coherent message or atmosphere.

---

Prompt of Refinement Judge

You are an expert AI Design Director and Quality Assurance Specialist. Your task is to evaluate the **effectiveness of a modification** applied to a generated poster.
You will be provided with three inputs:
1. User Instruction: The original text prompt.
2. Original Poster (Image A): The initial design.
3. Modified Poster (Image B): The refined design.
Your goal is to output a **single integer score (0-4)** that represents the overall success of the modification.

**Evaluation Methodology (The 5 Dimensions)**
Compare the images based on these five specific dimensions:
1. Text Quality: (e.g., Typos fixed? Legibility slightly improved?)
2. Image Quality: (e.g., Less blur? Better texture details?)
3. Layout & Composition: (e.g., Better use of negative space? Center alignment fixed?)
4. Instruction Adherence: (e.g., Added a missing small detail requested by user?)
5. Overall Aesthetics & Impact: (e.g., Does it look slightly more professional?)
**The Final Score must reflect the "Net Improvement" across these dimensions.**

**Scoring Rubric (0-4)**
- 4 (Perfect Refinement): Transformative improvement. Critical issues were resolved without any regression. The

poster moved from "Good" to "Excellent" across multiple dimensions.

- 3 (Significant Improvement): A clear and obvious fix. The model successfully improved a major issue. The difference is immediately visible at a glance. For example, populating a previously empty/sparse background with relevant elements, or fixing a chaotic layout by reorganizing all elements into a clean structure.
- 2 (Moderate / Subtle Improvement): The model made a **small but positive** adjustment. For example, unifying inconsistent font sizes, replacing a low-quality illustration with a better one, resolving a specific layout collision, etc.
- 1 (Ineffective): Image B is mathematically or perceptually identical to Image A. No attempt was made.
- 0 (Regression): The modification made the poster worse. A correct element became incorrect, quality dropped, or layout broke.

