# OpenReview forum: "PosterAgent: Agentic Poster Generation via Stage-Aware Reinforcement Learning"
_ICML.cc/2026/Conference — ICML 2026 regular_

### Official Review · Reviewer_gZy5 · 2026-03-10

**Soundness:** 3
**Presentation:** 3
**Significance:** 3
**Originality:** 3
**Overall Recommendation:** 5
**Confidence:** 3

**Summary:**

This paper proposes PosterAgent, a framework that reformulates automated poster generation as an iterative POMDP. Given a textual instruction, an MLLM (Qwen3-VL-8B) generates a JSON-based design specification, which is fed to a deterministic renderer that produces a poster image along with symbolic execution warnings. The key training contribution is Stage-Aware Reinforcement Learning (SARL), which separates the RL credit assignment into two phases: a draft phase that evaluates absolute initial quality, and a refinement phase that measures relative improvement from a fixed starting state. Experiments on an in-domain test set and the out-of-domain DESIGNERINTENTION benchmark show that an 8B model trained with this scheme achieves performance on par with GPT-5 and Gemini-2.5-Pro.

**Compliance With Llm Reviewing Policy:**

Affirmed.

**Final Justification:**

After Authors' rebuttal, my concers have been solved. I'll raise my score.

**Key Questions For Authors:**

Please see Weaknesses.

**Limitations:**

The authors did not include a discussion of the limitations of their work.

**Strengths And Weaknesses:**

Strengths

1. The SARL framework addresses a genuine and underappreciated failure mode in RL fine-tuning for multi-turn generation: outcome-only rewards conflate drafting quality with refinement quality, causing the policy to "collapse" by outputting identical designs across turns.

2. The POMDP formulation is principled and tight. The separation of visual observation (rendered image) and symbolic observation (execution warnings) is well-motivated: some layout errors (e.g., text overflow) are far easier to diagnose from structured feedback than from pixel-level inspection.

3. The out-of-domain results on DESIGNERINTENTION are impressive.

Weakness

1. The image generation in poster elements is fully delegated to Seedream 4.0, a closed-source proprietary model used within the renderer. This means the "Image Quality" scores in Tables 1 and 3 are largely measuring the quality of Seedream 4.0, not of PosterAgent's design decisions. The paper should either disentangle this dependency in the evaluation or explicitly acknowledge that image element quality is outside PosterAgent's scope.

2. The DESIGNERINTENTION benchmark is evaluated using GPT-4V (an older model), while PosterAgent-Test uses GPT-5. This inconsistency may affect the comparability of scores across benchmarks and slightly inflate the relative advantage on DESIGNERINTENTION

3. The fixed-horizon setting of T=3 is presented as a practical choice for "stable RL training and consistent evaluation," but the trade-off is never analyzed.

---

> ### Author Rebuttal · Authors · 2026-03-31
>
> We sincerely thank you for the thoughtful and constructive feedback. We are encouraged that you recognize the principled POMDP formulation, the genuine problem addressed by SARL, and the strong out-of-domain results. Below we address each weakness in detail.
>
> ---
>
> ## W1: Dependency on Seedream 4.0 for Image Generation
>
> We appreciate your observation and would like to clarify the scope of PosterAgent's design decisions regarding image elements.
>
> First, PosterAgent's control over image elements extends well beyond simply delegating to a T2I model. The agent must make several non-trivial decisions for each image element: (1) **composing an appropriate prompt** that is semantically aligned with the poster's theme, target audience, and visual tone specified in the user instruction; (2) **determining the bounding box** (position, size, and aspect ratio), which dictates how the image integrates with surrounding text and shapes in the overall composition. A well-generated image placed at the wrong size or position would still result in a poor poster. These decisions are learned through our training pipeline and are central to the agent's design capability.
>
> Second, this design follows the standard agentic paradigm, where the agent orchestrates external tools rather than reimplementing them—analogous to how SWE-Agent invokes code editors or how WebSailor uses web browsing tools. The framework is tool-agnostic: Seedream 4.0 can be replaced with any T2I model without architectural changes.
>
> Third, regarding evaluation, our "Image Quality" rubric (Appendix E.2) explicitly requires that visuals be *"highly relevant, powerfully enhance the poster's message, and perfectly match the style requested in the instruction"* to receive high scores. This means the score reflects the agent's design decisions (prompt composition, spatial placement, stylistic coherence) rather than solely the T2I model's generation fidelity.
>
> ---
>
> ## W2: Inconsistent Evaluation Models Across Benchmarks
>
> We would like to clarify that we **do not** compare absolute scores across Table 1 and Table 3—nor would such comparison be meaningful, as the two benchmarks adopt **entirely different evaluation dimensions**. Each benchmark is self-contained: within each table, all methods are evaluated by the same judge under the same rubric, so the relative rankings are strictly fair.
>
> The choice of GPT-4V for DESIGNERINTENTION was further motivated by two reasons: (1) to ensure fair comparison with prior work (COLE) under identical evaluation protocols, and (2) to demonstrate robustness—since our RL training uses GPT-5 as the judge, evaluating with a different model (GPT-4V) shows that PosterAgent's advantage is not an artifact of optimizing for a specific evaluator.
>
> Nonetheless, to fully address your concern, we conducted additional experiments evaluating all methods on DESIGNERINTENTION using GPT-5 as the judge:
>
> | Method | Design | Content | Typo. | Graphic | Innov. | Overall |
> |---|---|---|---|---|---|---|
> | **PosterAgent** | **7.620** | **7.886** | **7.190** | **8.046** | **5.584** | **7.265** |
> | GPT-5 | 7.286 | 7.818 | 6.798 | 7.872 | 5.418 | 7.038 |
> | Seedream 4.0 | 6.984 | 7.540 | 6.462 | 7.636 | 5.120 | 6.748 |
>
>
> PosterAgent maintains a clear lead over all baselines under GPT-5 evaluation, confirming that our conclusions are robust to the choice of evaluator. We will include these results in the revision.
>
> ---
>
> ## W3: Analysis of the Fixed-Horizon Trade-off (T=3)
>
> We thank you for raising this point. We note that this trade-off is analyzed in Section 4.5 ("Impact of Refinement Steps") and Figure 3, which plots overall score as a function of T from 0 to 4. The results show that performance improves progressively from T=0 (6.80) to T=3 (7.41), with T=4 yielding no further gain (7.39).
>
> The diminishing returns beyond T=3 arise because most correctable layout and typography issues are resolved within the first few rounds. Additional iterations risk over-editing—e.g., introducing unnecessary elements that disrupt an already well-organized layout. Meanwhile, each refinement round adds approximately 5.2k tokens to the context (2.8k visual tokens from the rendered poster + 2.4k from the model's response), scaling linearly with T. T=3 thus represents a practical sweet spot balancing quality gains against computational overhead.
>
> ---
>
> Thank you again for evaluating our manuscript. Please let us know if you have any further questions, as we are happy to continue the discussion. If you find that our response addresses your concerns, would you kindly consider raising your rating score for our paper? We greatly appreciate your consideration.

---

> > ### Author Rebuttal · Reviewer_gZy5 · 2026-04-03
> >
> > Thanks for Authors' rebuttal. My concers have been solved. I'll keep positive score.

---

### Official Review · Reviewer_nT1B · 2026-03-12

**Soundness:** 3
**Presentation:** 3
**Significance:** 2
**Originality:** 2
**Overall Recommendation:** 4
**Confidence:** 4

**Summary:**

The paper proposes PosterAgent, which formulates poster generation as an agentic workflow with an initial draft followed by multi-turn self-refinement. The system uses a structured JSON design representation, a renderer, and MLLM-based feedback over rendered outputs and warnings, while the proposed SARL training scheme separates draft and refinement rewards to improve credit assignment. Experiments on PosterAgent-Test and DESIGNERINTENTION show stronger results than T2I models, and comparable results with frontier models.

**Compliance With Llm Reviewing Policy:**

Affirmed.

**Final Justification:**

The new evaluation results and qualitative examples addressed my concerns.
Nevertheless, I still find the technical contributions (learning multi-turn refinement via RL, stage-aware rollout and rewards) is somewhat limited, and the specific focus on poster generation restricts the work’s overall significance.

**Key Questions For Authors:**

Can the authors provide more challenging qualitative examples showing that the RL method is capable of improving complex poster layouts beyond basic text-heavy ones?

**Limitations:**

See weaknesses

**Strengths And Weaknesses:**

**Strengths**

- The motivation is reasonable. Poster design is naturally iterative, so modeling it as draft + multi-turn refinement makes sense.
- The overall system design is practical: structured design output, rendering, and feedback-driven revision fit the task well.
- The experimental section is fairly complete and gives decent support for the proposed framework.
- Training agentic models for poster/graphic design are somewhat novel.

**Weaknesses**
- The reward judge during training and the in-domain automatic evaluator are both GPT-5. This raises a risk of evaluator overfitting or reward hacking.
- I am also not fully convinced by most of the novelty claims.
  - Prior work is described as mostly one-shot and lacking iterative repair, but this is not fully accurate. For example, BannerAgency (arXiv:2503.11060) already models graphic design as a multi-turn visual-feedback agent workflow. Paper2Poster (arXiv:2505.21497) also explores iterative refinement with visual feedback for poster generation.
  - Turn-level rewards are already common in agentic RL settings, and the Group-Normalized Advantages formulation appears very close to GRPO-style RL. And the connections are under-cited.
- The qualitative examples look fairly simple: vertical, text-heavy, linear layouts with limited stylistic diversity. It is hard to tell whether the method really helps with more complex poster design.

---

> ### Author Rebuttal · Authors · 2026-03-31
>
> We sincerely thank you for the thorough review. We are encouraged that you recognize the reasonable motivation of our iterative design formulation, the practical system design, the fairly complete experiments, and the novelty of training agentic models for poster design. We address each concern below.
>
> ---
>
> ## W1: Risk of Evaluator Overfitting / Reward Hacking
>
> We provide three lines of evidence against this concern.
>
> **Out-of-domain generalization.** DESIGNERINTENTION uses GPT-4V (not GPT-5) as evaluator, with different prompt styles, dimensions, and artifact types from our training distribution. PosterAgent still achieves the best overall score (7.66), ruling out overfitting to the training-time judge.
>
> **Human evaluation.** As shown in Figure 2, human preferences align with automatic evaluation trends. PosterAgent wins against GPT-5, Seedream 4.0, and Qwen-Image on both benchmarks, confirming genuine quality gains rather than reward hacking.
>
> **Cross-evaluator validation.** We additionally evaluate on PosterAgent-Test using **Gemini-2.5-Pro**, entirely unseen during training:
>
> | Method | Text | Image | Layout | Instr. | Aesth. | Overall |
> |---|---|---|---|---|---|---|
> | PosterAgent | **8.15** | 8.01 | **7.34** | **7.31** | **7.48** | **7.66** |
> | GPT-5 | 7.22 | **8.05** | 7.30 | 7.66 | 7.24 | 7.49 |
> | Seedream 4.0 | 4.68 | 7.51 | 5.99 | 6.01 | 5.31 | 5.90 |
>
> The ranking is **fully consistent** with the GPT-5 evaluation: PosterAgent remains competitive with GPT-5 and substantially outperforms Seedream 4.0, which further excludes evaluator-specific overfitting. We will include these results in the revision.
>
> ---
>
> ## W2: Novelty Concerns
>
> **Distinction from prior iterative work.** We thank you for pointing out BannerAgency and Paper2Poster and will update our related work and novelty claims accordingly in the revision. We acknowledge that our original characterization of prior work as lacking iterative refinement was not fully accurate. However, a critical distinction remains: both methods employ existing foundation models as agents **without any training** to improve refinement capability. Our core contribution is not the iterative workflow itself, but **how to effectively train** multi-turn refinement via RL.
>
> This is empirically validated. Qwen3-VL-8B (untrained backbone) already operates within our full iterative workflow—same schema, prompts, and multi-turn protocol—yet scores only 5.08. After SFT, the score rises to 6.76; after SARL, it reaches 7.41 (+0.65). This demonstrates that RL-trained refinement ability, not the workflow alone, drives performance.
>
> **Turn-level rewards and Group-Normalized Advantages.** We acknowledge existing work on turn-level rewards in agentic RL and will add citations. Our intent is not to claim that assigning non-terminal rewards is new in general. The key difference of SARL is the **stage-aware rollout construction**: instead of assigning different rewards to turns within a single rollout, we explicitly decouple drafting and refinement at the sampling level. In particular, refinement trajectories are branched from the same fixed seed draft, and are scored by relative improvement against that shared starting point. This controls for initial-draft variance and isolates the marginal contribution of refinement actions, which standard within-trajectory turn-level reward formulations do not directly target.
>
> Regarding Group-Normalized Advantages, we agree this is widely adopted (as in GRPO) and do not claim it as a contribution. We will add the GRPO citation.
>
> ---
>
> ## W3: Layout Diversity and Complexity
>
> We provide additional qualitative examples here (https://anonymous.4open.science/r/ICML-PosterAgent/Supplementary%20Case%20Study.md). When given user instructions that specify richer structural requirements, PosterAgent produces designs with multi-column layouts, parallel information streams, zigzag spatial arrangements, and dense compositions mixing heterogeneous elements (annotated diagrams, maps, data panels, icon grids, timelines) — well beyond simple vertical text-heavy layouts.
>
> These cases also highlight the limitations of baselines. GPT-5 frequently suffers from typographic issues such as undersized text and improper element overlap, resulting in cluttered and hard-to-read compositions. Seedream 4.0 struggles more fundamentally with **content planning** and **accurate text rendering**, often producing garbled or incorrect text. In contrast, PosterAgent's iterative self-refinement loop enables it to detect and correct such layout and rendering flaws, consistently yielding well-organized, legible designs across varying levels of complexity.
>
> ---
>
> Thank you again for evaluating our manuscript. Please let us know if you have any further questions, as we are happy to continue the discussion. If you find that our response addresses your concerns, would you kindly consider raising your rating score for our paper? We greatly appreciate your consideration.

---

> > ### Author Rebuttal · Reviewer_nT1B · 2026-04-03
> >
> > I thank the authors for the rebuttal. The new evaluation results and qualitative examples addressed my concerns. I am raising the score to 4.
> >
> > Nevertheless, I still find the technical contributions (learning multi-turn refinement via RL, stage-aware rollout and rewards) is somewhat limited, and the specific focus on poster generation restricts the work’s overall significance.

---

> > > ### Author Response · Authors · 2026-04-03
> > >
> > > We sincerely thank you for the thoughtful follow-up and for acknowledging that our new results and examples addressed your earlier concerns.
> > >
> > > Regarding the remaining points: we agree that the technical contributions of SARL are focused rather than sweeping, but we believe this focus is a strength — the stage-aware decoupling addresses a concrete and previously unsolved credit-assignment problem in multi-turn RL for generative agents, and our ablations (overall score: 6.95 → 7.41) confirm it is not a trivial design choice. We also note that the training methodology itself (stage-aware rollout construction, relative-improvement scoring from a shared seed state) is general and not restricted to poster generation.
> > >
> > > On the concern about task-specific scope: we view poster generation as a representative instance of a broader class of structured visual design tasks. From a task-structure perspective, presentation design, web/UI design, and document layout all share the same core loop — understanding requirements, producing an editable structured representation, and iteratively refining it based on visual feedback. PosterAgent's framework and training paradigm are directly transferable to these settings, and we see extending to such tasks as a natural and promising direction for future work.
> > >
> > > Finally, we noticed that the updated score may not yet be reflected in the system. Could we kindly ask you to verify that the score has been updated accordingly? We truly appreciate your time and constructive engagement throughout this discussion.

---

### Official Review · Reviewer_uGQ2 · 2026-03-12

**Soundness:** 2
**Presentation:** 3
**Significance:** 3
**Originality:** 3
**Overall Recommendation:** 4
**Confidence:** 3

**Summary:**

This paper presents PosterAgent, a poster-generation system that treats design as a multi-step agent problem rather than a one-shot generation task. The model first produces a structured JSON poster draft, renders it, inspects the rendered result, and then refines the design over several turns. The main technical idea is Stage-Aware Reinforcement Learning (SARL), which separates learning for initial drafting from learning for later refinements. Concretely, the method scores multiple independent initial designs with an absolute-quality judge, then scores multiple refinement trajectories from the same fixed seed state with a relative-improvement judge, and uses stage-specific normalized advantages for policy updates. The reported results show large gains over the base backbone, clear improvements from iterative refinement and stage-aware rewards, and competitive performance relative to strong proprietary models in both automatic and human evaluation.

**Compliance With Llm Reviewing Policy:**

Affirmed.

**Final Justification:**

I have considered the authors’ rebuttal carefully. The response addresses the main questions I had and provides helpful clarification on several technical points. While some minor limitations remain, they do not substantially affect my overall assessment. I therefore keep my original recommendation unchanged.

**Key Questions For Authors:**

1. The main quantitative claims rely on GPT-5 and GPT-4V judges, and the reported differences versus baselines are very small. Can you provide significance testing, judge agreement analysis, or stronger human evaluation to show that these gains are real and stable?

2. How sensitive is performance to the number of refinement steps at test time? Please show quality versus number of iterations, along with inference cost. This would help clarify whether the gains come from a small number of useful edits or from simply giving the model more chances.

3. How robust is the method to different judge models during training and evaluation? Since the paper relies heavily on MLLM judges, it would be useful to know whether the conclusions hold under different evaluators, or whether the training reward is tightly coupled to the specific judge used.

4. What are the most common failure cases after refinement is complete? A dedicated failure analysis would help readers understand the current limits of the approach and would make the paper feel more balanced and trustworthy.

**Limitations:**

No.

The paper would be stronger with a fuller discussion of a few points: its reliance on proprietary models for data generation, evaluation, and image rendering; the possibility of bias or instability in MLLM-based judging; the risk that such a system could be used to create misleading or overly persuasive posters; and the difference between strong benchmark scores and actual usefulness in real design workflows.

**Strengths And Weaknesses:**

1. Soundness:
The paper has a clear technical idea and a method that fits the task well. Treating poster design as an iterative agent problem is reasonable, and the stage-aware RL setup is a sensible way to separate first-draft quality from later refinement quality. The ablations are useful and mostly support the main claims: self-refinement helps, stage-aware rewards help, and renderer warnings also matter.
The main weakness is the evidence base. Much of the evaluation depends on MLLM judges, including GPT-5 and GPT-4V, and the strongest headline comparisons are based on very small margins. I did not see significance testing, confidence intervals, or strong agreement analysis to show that these gains are stable.

2. Presentation:
The paper is generally well organized and easy to follow. The method section is structured well, and the figures make the training and inference process understandable. The narrative from task formulation to training setup to evaluation is coherent.

3. Significance:
The paper targets a useful and relevant problem. Poster generation is not just an aesthetic task; it also requires layout, text handling, hierarchy, and editability. A structured agentic approach could be valuable for real design workflows, and the idea of learning how to draft and revise in stages could be useful beyond posters as well. However, the current paper stops short of showing broader practical impact. Most of the evidence is benchmark-based and judge-based, rather than grounded in real user studies or real design workflows. There is little evidence yet that the system saves time, improves usability, or works reliably in realistic settings with actual users or designers.

4. Originality:
The paper has a real novelty angle. The individual pieces are familiar, but the combination is thoughtful and well tailored to the task. The most original part is the stage-aware RL design that separates drafting from refinement and uses different grouping and normalization strategies for the two stages. It builds on existing ideas from agentic refinement, RL with learned judges, and structured generation. So the originality comes more from the task-specific formulation and training design than from a fundamentally new modeling approach.

---

> ### Author Rebuttal · Authors · 2026-03-31
>
> We sincerely thank you for the thorough and constructive review. We appreciate your recognition of our technical contributions, the clarity of our presentation, and the potential significance of the agentic design paradigm. Below we address each concern in detail.
>
> ---
>
> ## W1: Statistical Significance and Judge-human Agreement
>
> We respectfully note that the improvements over most baselines are **substantial**. On PosterAgent-Test (9-point scale), PosterAgent outperforms Seedream 4.0 by 1.13 and the untrained Qwen3-VL-8B by 2.33 — corresponding to 12.6% and 25.9% relative gains. We now provide paired bootstrap 95% confidence intervals for the mean score difference (PosterAgent − Baseline):
>
> | Baseline | Mean Δ | 95% CI |
> |---|---|---|
> | Seedream 4.0 | +1.13 | [0.926, 1.323] |
> | Qwen3-VL-8B | +2.33 | [2.075, 2.596] |
> | PosterAgent (Cold-Start) | +0.64 | [0.452, 0.858] |
> | GPT-5 | −0.02 | [−0.257, 0.233] |
>
> The CIs for the first three comparisons exclude zero, confirming statistically significant improvements. The CI for GPT-5 contains zero, which is consistent with our claim of *comparable* — not superior — performance.
>
> To validate judge-human alignment, we encode human GSB (Good/Same/Bad) labels as +1/0/−1 and compute rank correlations against MLLM judge score differences:
>
> | Judge Model (Benchmark) | Spearman ρ | Kendall τ |
> |---|---|---|
> | GPT-5 (PosterAgent-Test) | 0.602 | 0.497 |
> | GPT-4V (DesignerIntention) | 0.542 | 0.473 |
>
> Given the inherent subjectivity of design evaluation, these correlations are strong and confirm that the judge-based rewards are well aligned with human preference.
>
> ---
> ## W2: Quality vs. Refinement Steps at Test Time with Cost
>
> We evaluate the *same* model (trained with T=3) at varying inference-time refinement steps T=0,1,2,3, using GPT-5 as the judge on PosterAgent-Test:
>
> | Step | Text | Image | Layout | Instr. | Aesth. | Overall |
> |---|---|---|---|---|---|---|
> | T=0 | 6.74 | 7.04 | 6.82 | 6.66 | 6.67 | 6.78 |
> | T=1 | 6.89 | 7.17 | 7.01 | 6.87 | 6.82 | 6.95 |
> | T=2 | 7.17 | 7.39 | 7.26 | 7.14 | 7.12 | 7.22 |
> | T=3 | 7.32 | 7.60 | 7.44 | 7.38 | 7.29 | 7.41 |
>
> Quality improves monotonically at every step and across all dimensions, confirming that the gains stem from meaningful edits rather than random variation. Regarding inference cost, each refinement round adds approximately 5.2k tokens to the context (2.8k visual tokens from the rendered poster + 2.4k from the model's response), scaling linearly with T. This is a modest overhead given the consistent quality gains.
>
> ---
>
> ## W3: Robustness to Different Judge Models
>
> Following your suggestion, we additionally evaluate on PosterAgent-Test using **Gemini-2.5-Pro** as the judge, a model **entirely unseen** during training:
>
> | Method | Text | Image | Layout | Instr. | Aesth. | Overall |
> |---|---|---|---|---|---|---|
> | PosterAgent | **8.15** | 8.01 | **7.34** | **7.31** | **7.48** | **7.66** |
> | GPT-5 | 7.22 | **8.05** | 7.30 | 7.66 | 7.24 | 7.49 |
> | Seedream 4.0 | 4.68 | 7.51 | 5.99 | 6.01 | 5.31 | 5.90 |
>
> The ranking is **fully consistent** with the GPT-5 evaluation: PosterAgent remains competitive with GPT-5 and substantially outperforms Seedream 4.0. This confirms that our conclusions are not artifacts of the specific judge used during training. We acknowledge that training with alternative judges is an interesting direction and plan to include such experiments in a future version.
>
> ---
>
> ## W4: Common Failure Cases
>
> We identify two recurring failure modes after refinement:
>
> 1. Suboptimal use of space. When the user instruction specifies relatively little content, the agent occasionally leaves some underutilized regions in the layout. While the overall composition remains readable, a human designer would typically fill such areas with decorative elements or adjust spacing more aggressively to achieve a tighter visual balance.
>
> 2. Content hallucination. As an 8B-parameter model, the backbone occasionally generates factually inaccurate details, a known limitation of smaller LLMs that is orthogonal to the layout and refinement capabilities of our framework.
>
> We have included illustrative failure examples here (https://anonymous.4open.science/r/ICML-PosterAgent/Failure%20Cases.md) to provide a balanced view of the system's current limitations.
>
> ---
>
> Thank you again for evaluating our manuscript. Please let us know if you have any further questions, as we are happy to continue the discussion. If you find that our response addresses your concerns, would you kindly consider raising your rating score for our paper? We greatly appreciate your consideration.

---

> > ### Author Rebuttal · Reviewer_uGQ2 · 2026-04-03
> >
> > Thanks for the rebuttal. It addresses most of my main concerns.
> >
> > On the positive side, the added confidence intervals are useful and do address my concern about whether the larger gains over Seedream 4.0, the base Qwen3-VL-8B, and the cold-start variant are statistically stable. The clarification that GPT-5 is only comparable rather than exceeded is appropriate. The added judge–human correlation analysis is also a constructive addition, though I would describe the alignment as moderate rather than fully convincing given the centrality of judge-based evaluation. The response on evaluator robustness is directionally reassuring as well, since the ranking appears consistent under Gemini-2.5-Pro.
> >
> > However, I still think the paper has some remaining limitations, especially its reliance on proprietary judge models and the relatively light failure analysis. So I would keep my score at weak accept. The paper is technically solid, but the evidence is still too judge-heavy and not yet strong enough on practical validation to justify a higher rating.

---

### Official Review · Reviewer_rYr5 · 2026-03-12

**Soundness:** 2
**Presentation:** 2
**Significance:** 1
**Originality:** 2
**Overall Recommendation:** 3
**Confidence:** 3

**Summary:**

This paper proposes an agentic framework for poster generation based on an editable JSON representation and iterative self-refinement.
It also introduces stage-aware RL (SARL) to separately optimize drafting and refinement.
Experiments show fair empirical performance (though the paper could better clarify its novelty relative to recent structured generation methods and discuss the efficiency cost of iterative refinement).

**Compliance With Llm Reviewing Policy:**

Affirmed.

**Final Justification:**

I thank the authors for the rebuttal and additional results.
The paper is clear and technically reasonable, and the rebuttal addressed some empirical concerns.
However, from a novelty perspective, I would like to maintain my original assessment.

**Key Questions For Authors:**

Please see the weaknesses below.

**Limitations:**

Some of the limitations could be mitigated by providing additional experiments, especially on comparisons with related structured generation methods and on the cost-quality trade-off of iterative refinement.

**Strengths And Weaknesses:**

Strengths and Weaknesses

S1. The use of an editable structured representation (JSON) for poster generation is well motivated.
It provides a practical interface for iterative refinement and makes the generation process more controllable (than direct image synthesis).

S2. The paper’s attempt to separately optimize drafting and refinement is a meaningful contribution.
Distinguishing initial generation quality from revision quality is intuitively sound and aligns well with the iterative nature of design.

W1. The overall pipeline appears closely related to recent work on paper-to-poster and paper-to-web generation, especially in its use of structured intermediate representations (IR) and iterative self-refinement after rendering. While the target domain is different, the paper would benefit from a clearer discussion of how its novelty goes beyond a domain shift and from a stronger comparison to these recent studies.

* Paper2Poster: Towards Multimodal Poster Automation from Scientific Papers (NeurIPS 2025 D&B)
* P2P: Automated Paper-to-Poster Generation and Fine-Grained Benchmark (ICLR 2026)
* PosterForest: Hierarchical Multi-Agent Collaboration for Scientific Poster Generation (arXiv 2025.08)
* Paper2Web: Let's Make Your Paper Alive! (arXiv 2025.10)

W2. The RL framework relies on an MLLM-as-a-Judge for reward construction, so the quality of the learned policy may depend heavily on the reliability of that judge.
It would strengthen the paper to provide more evidence that the judge-based reward is well aligned with human preference and actual design quality.
(Since the MLLM judges themselves may not be good at this task.)
Additionally, it is necessary to check whether using the MLM judge for learning and then using it again for eval is not a circular logic error.

W3. The evaluation is centered mainly on general-purpose frontier models (GPT, Qwen, Seedream), while comparisons with prior layout-driven or structured poster generation methods are relatively limited.
More direct qualitative and quantitative comparisons with methods such as COLE and OpenCOLE would better clarify the contribution of the proposed framework.
In addition, since the model is built on Qwen3-VL-8B, comparison with the backbone itself would help isolate the benefit of the proposed method.

W4. The qualitative comparison in Figure 4 is not fully convincing as evidence of a clear visual advantage.
While PosterAgent appears competitive, the differences from some baselines are not sufficiently pronounced to clearly demonstrate the unique benefit of the proposed agentic refinement framework.

W5. Figure 6 is useful in illustrating the iterative refinement process, but the visual changes appear relatively modest and seem to focus mainly on local text or formatting adjustments.
As a result, the example does not fully demonstrate whether the method can perform more substantial layout-level reorganization during refinement.

W6. The proposed iterative refinement framework likely incurs nontrivial additional inference cost, but the paper does not analyze the trade-off between quality improvement and efficiency (e.g., token usage, latency, or rendering overhead).
Such analysis would strengthen the practical value of the work.

---

> ### Author Rebuttal · Authors · 2026-03-31
>
> We sincerely thank you for the thorough and constructive evaluation. We appreciate your recognition of the structured JSON representation and the stage-aware optimization design. Below we address each concern.
>
> ---
>
> ## W1: Relation to Paper-to-Poster and Paper-to-Web Work
>
> Thank you for raising these relevant studies. We will cite and discuss them. We acknowledge the shared high-level idea of structured IRs with iterative refinement. Our novelty, however, is **not** a domain shift but lies in **how the agent is trained to refine**.
>
> **SARL as the core contribution.** All cited works rely on prompting, SFT, or multi-agent collaboration. None employ RL-based training, nor address the credit-assignment problem when a single model must both draft and refine. SARL is the first paradigm that decouples these two stages via stage-specific rollout sampling and reward shaping, enabling principled optimization of each stage's distinct objective.
>
> **Additionally, a broader task scope.** The cited works take a structured scientific paper as input, with pipelines built around document parsing and summarization. PosterAgent accepts open-ended natural-language instructions across eight domains, requiring the agent to plan information hierarchy and generate content from scratch.
>
> ---
>
> ## W2: MLLM-as-Judge Reliability and Circular Evaluation
>
> **Circular evaluation is avoided.** Our training judge is GPT-5, while DESIGNERINTENTION uses GPT-4V following the original COLE paper. Beyond the different judge models, the two benchmarks adopt **entirely different evaluation dimensions** (e.g., text/image/layout/instruction/aesthetics vs. design/content/typography/graphic/innovation). Crucially, we conduct **human evaluation** on both benchmarks (Figure 2), providing an entirely independent assessment.
>
> **Judge–human alignment.** Following your suggestion, we computed Spearman ρ and Kendall τ between MLLM judge score differences and human preferences (+1/0/−1):
>
> | Judge Model | ρ | τ |
> |---|---|---|
> | GPT-5 (PosterAgent-Test) | 0.602 | 0.497 |
> | GPT-4V (DESIGNERINTENTION) | 0.542 | 0.473 |
>
> Given the inherent subjectivity of design evaluation, these correlations are strong and confirm that the judge-based rewards are well aligned with human preference.
>
> ---
>
> ## W3: Comparison with Layout-Driven Methods and Backbone
>
> We respectfully note that these comparisons are already present. Table 3 includes COLE and OpenCOLE on DESIGNERINTENTION (PosterAgent: 7.66 vs. 6.00 and 6.30). Tables 1 and 3 report the Qwen3-VL-8B backbone: overall improves from 5.08→7.41 (PosterAgent-Test) and 6.36→7.66 (DESIGNERINTENTION), isolating the benefit of our framework. We will make these comparisons more prominent.
>
> ---
>
> ## W4 & W5: Qualitative Evidence
>
> **W4 (Comparison with baselines).** We address this in our response to Reviewer nT1B (W3), where we provide additional complex cases and analyze specific baseline limitations. We kindly refer you there to avoid redundancy.
>
> **W5 (Scope of refinement).** The incremental nature of refinement is by design. The draft stage establishes global layout and information hierarchy, while refinement is explicitly scoped for targeted improvements—spacing, font consistency, alignment, and element-level polish. This mirrors professional design practice, where major structural decisions are made early and iterations focus on fine-tuning.
>
> ---
>
> ## W6: Trade-off between Quality Improvement and Efficiency
>
> **Training-time analysis (Figure 3).** We trained separate models with refinement horizons T∈{0,1,2,3,4}. Quality improves from T=0 to T=3 (6.80→7.41) and plateaus at T=4, showing three rounds strike the best quality–cost balance.
>
> **Inference-time analysis.** We examined the model trained with T=3 by evaluating intermediate outputs at each step:
>
> | Step | Text | Image | Layout | Instr. | Aesth. | Overall |
> |---|---|---|---|---|---|---|
> | 0 | 6.74 | 7.04 | 6.82 | 6.66 | 6.67 | 6.78 |
> | 1 | 6.89 | 7.17 | 7.01 | 6.87 | 6.82 | 6.95 |
> | 2 | 7.17 | 7.39 | 7.26 | 7.14 | 7.12 | 7.22 |
> | 3 | 7.32 | 7.60 | 7.44 | 7.38 | 7.29 | 7.41 |
>
> Every step yields consistent gains, so users can early-stop at any point for a quality–latency tradeoff.
>
> **Token cost.** Each refinement round adds ~5.2k context tokens (2.8k visual + 2.4k response), growing roughly linearly. This overhead is modest relative to the quality gain. We will include this analysis in the revision.
>
> ---
>
> Thank you again for evaluating our manuscript. Please let us know if you have any further questions, as we are happy to continue the discussion. If you find that our response addresses your concerns, would you kindly consider raising your rating score for our paper? We greatly appreciate your consideration.

---

> > ### Author Rebuttal · Reviewer_rYr5 · 2026-04-05
> >
> > Thank you for the rebuttal and additional clarifications.
> > I appreciate the effort to address the concerns, and your response has helped clarify several aspects of the work.
> > However, a few concerns remain:
> >
> > 1. From the results presented in Figures 4, 5, and 6, as well as the response to Reviewer nT1B’s W3, it is still difficult to confirm a clear and significant advantage of the proposed method over existing baselines (in the aspect of qualitative evaluation). Could you tell me in which areas there was a clear and consistent gain?
> >
> > 2. I understand that this work effectively applies a GRPO-style approach to the poster generation domain. However, due to the lack of a detailed ablation study on how each specific reward affects the generation, it remains difficult to interpret the direct impact of individual rewards. While Supplementary Section C shows the "valid refinement steps," providing qualitative examples of how the actual visual outputs change based on different reward settings would significantly strengthen the paper.
> >
> > 3. Overall, the study appears to focus on incorporating RL-based engineering elements into existing poster generation frameworks.
> >
> > Addressing these remaining points would further improve the quality of the paper. For now, I will maintain my current score.

---

> > > ### Author Response · Authors · 2026-04-08
> > >
> > > Thank you for the continued discussion. We address the remaining points below.
> > >
> > > ---
> > >
> > > ## Point 1: Specific qualitative advantages.
> > >
> > > PosterAgent shows consistent gains in three areas: (1) text rendering accuracy (vs. Seedream 4.0's pervasive garbled text), (2) layout consistency and balance (vs. GPT-5's uneven element sizing), and (3) content organization without duplication or omission (vs. both baselines). These advantages are visible in the additional case studies we provided in our response to Reviewer nT1B’s W3, where the comparison with baseline models more clearly illustrates PosterAgent’s strengths along these dimensions.
> > >
> > > ---
> > >
> > > ## Point 2: Ablation on individual reward components.
> > >
> > > This is a valuable suggestion. While our current ablation (Table 2, "w/o stage-aware reward") demonstrates the aggregate benefit of decoupled rewards, we agree that showing how individual reward dimensions (e.g., text quality, layout, instruction adherence) influence the generated output would provide deeper insight. We will include a per-dimension reward ablation with qualitative examples in the revised version.
> > >
> > > ---
> > >
> > > ## Point 3: Characterization as "RL engineering on existing frameworks."
> > >
> > > We respectfully disagree with this characterization. SARL addresses a fundamental credit-assignment problem in multi-turn generative tasks: when a single policy must both draft and refine, a naive outcome-only reward conflates the two behaviors, leading to refinement collapse (as evidenced by Figure 7). The stage-aware decoupling — separate sampling phases, stage-specific rewards, and independent advantage normalization — is a principled solution to this problem, not a domain-specific engineering choice.
> > >
> > > More importantly, we view SARL as a general paradigm for training agentic systems that operate through iterative creation and revision. The draft-then-refine workflow is not unique to poster generation — it naturally arises in presentation creation, web page design, UI/UX prototyping, and other structured design tasks where an agent must first produce a global plan and then iteratively polish the output based on rendered feedback. SARL's stage-aware credit assignment is directly applicable to any such setting. We believe this generalizability is a key contribution that extends well beyond the poster domain explored in this paper.

---

### Decision · Program_Chairs · 2026-04-30

**Decision:**

Accept (regular)

**Comment:**

This paper receives final ratings of (3, 4, 4, 5). The reviewers appreciate that the paper introduces a principled Stage-Aware Reinforcement Learning (SARL) framework that effectively addresses the credit-assignment problem in multi-turn poster generation. The reviewers agree that the approach is effective, particularly in its ability to separate initial drafting quality from iterative refinement quality. The AC agrees with the strengths and an acceptance is recommended.